# UniF²ace: A Unified Fine-grained Face Understanding and Generation Model

**Junzhe Li**[1,2][*], **Sifan Zhou**[3][*], **Liya Guo**[4], **Xuerui Qiu**[5,12], **Linrui Xu**[6], **Delin Qu**[7],
**Tingting Long**[2], **Chun Fan**[2], **Hehe Fan**[8], **Jun Liu**[9], **Ming Li**[10][†] **Shuicheng Yan**[11],

[1]School of Computer Science, Peking University, [2]Computer Center, Peking University,
[3]Carnegie Mellon University [4]Tsinghua University
[5]Institute of Automation, Chinese Academy of Sciences [6]Central South University
[7]Fudan University [8]Zhejiang University [9]Lancaster University
[10]Guangdong Laboratory of Artificial Intelligence and Digital Economy (SZ)
[11]National University of Singapore [12]Zhongguancun Academy

## Abstract

Unified multimodal models (UMMs) have emerged as a powerful paradigm in fundamental cross-modality research, demonstrating significant potential in both image understanding and generation. However, existing research in the face domain primarily faces two challenges: **(1) fragmentation development**, with existing methods failing to unify understanding and generation into a single one, hindering the way to artificial general intelligence. **(2) lack of fine-grained facial attributes**, which are crucial for high-fidelity applications. To handle those issues, *we propose UniF²ace, the first UMM specifically tailored for* fine-grained *face understanding and generation*. **First**, we introduce a novel theoretical framework with a Dual Discrete Diffusion (D3Diff) loss, unifying masked generative models with discrete score matching diffusion and leading to a more precise approximation of the negative log-likelihood. Moreover, this D3Diff significantly enhances the model's ability to synthesize high-fidelity facial details aligned with text input. **Second**, we propose a multi-level grouped Mixture-of-Experts architecture, adaptively incorporating the semantic and identity facial embeddings to complement the attribute forgotten phenomenon in representation evolvement. **Finally**, to this end, we construct UniF²aceD-1M, a large-scale dataset comprising *130K* fine-grained image-caption pairs and *1M* visual question-answering pairs, spanning a much wider range of facial attributes than existing datasets. Extensive experiments demonstrate that UniF²ace outperforms existing models with a similar scale in both understanding and generation tasks, with 7.1% higher Desc-GPT and 6.6% higher VQA-score, respectively. Codes and datasets are available at UniF²ace.

## 1 Introduction

Recently, unified multimodal models (UMMs) have emerged as a vibrant research area enabling cross-modality understanding and generation within a single "any-to-any" framework, marking a significant step toward artificial general intelligence (AGI) (Wu et al., 2024a; Shi et al., 2024; Li et al., 2024a; Zhou et al., 2024; Zhou, 2025; Team, 2024; Xie et al., 2024; 2025; Zhang et al., 2025; Hu et al., 2025; Chen et al., 2024a; 2025a;b; Liu et al., 2025a;b; 2026; Liu et al.; Ni et al., 2025; Yu et al., 2025). Given the central role of faces in daily life, applying this unified paradigm to achieve fine-grained face understanding and generation is essential for developing human-centric AGI. The practical applications are vast and critical: accurate face understanding is pivotal for identity verification (Srinivasan et al., 2024; Roshdy et al., 2024) and human-computer interaction (Liu, 2024; Chowdary et al., 2023), while high-fidelity face generation drives progress in creative industries (Melnik et al., 2024), virtual avatars (Yan et al., 2024), and data augmentation for model robustness (Melzi et al., 2023). These

---

[*]Equal contribution. (lijunzhe1028@stu.pku.edu.cn)
[†]Corresponding author.

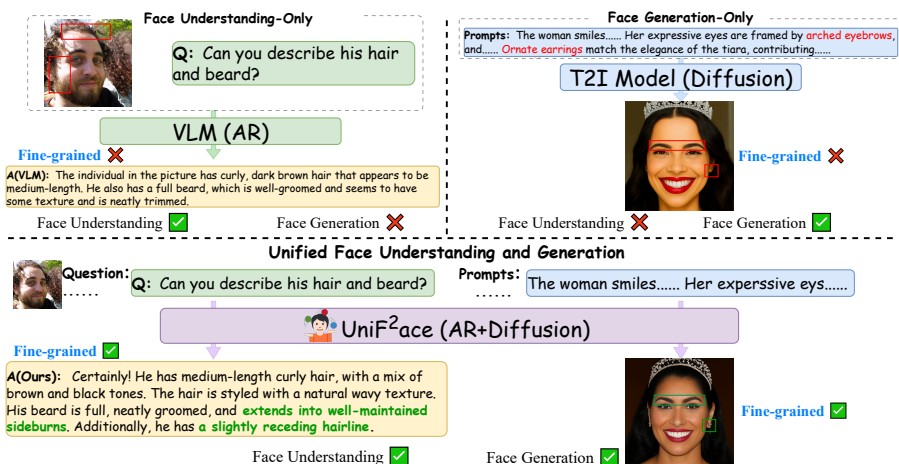

Figure 1: UniF²ace is the first unified multimodal model designed for face understanding and generation, encompassing tasks such as visual question answering(VQA) and text-to-image generation. The generated responses and images demonstrate UniF²ace's potential in fine-grained face attributes.

demanding real-world needs urge facial research to push the boundaries of multimodal understanding and generative modeling.

As shown in Fig. 1, despite the critical importance of human faces, existing research faces two fundamental and intertwined challenges: **First, the field remains fragmented**, with current methodologies treating face understanding and generation as separate endeavors, failing to unify these capabilities into a single framework. Typically, face understanding models are often based on fine-tune pretrained multimodal large language models (MLLMs) on facial images with coarse text descriptions (Chettaoui et al., 2025; Sun et al., 2024a; Xing et al., 2024). Face generation models (Huang et al., 2023; Nair et al., 2023; Kim et al., 2024) often incorporate visual information, such as semantic masks and sketches, to guide high-fidelity face synthesis, but they cannot achieve direct generation from detailed captions to faces. This leads to disjointed workflows that are both computationally inefficient and functionally restrictive. Crucially, the absence of a unified framework represents a significant hurdle towards the realization of AGI within the domain of human faces. **Second, there is a pervasive lack of fine-grained information processing** across both understanding and generation tasks. This challenge stems from three problems: (a) The discrete diffusion model inherits the advantages of diffusion for image generation while enabling better scalability modeling with text tokens in UMMs (Yang et al., 2025; Xie et al., 2024). However, its specific implementation primarily relies on masked generative models (Chang et al., 2022), lacking a combination with accurate score matching (Lou et al., 2024), making it challenging to generate precise image details; (b) detailed attribute representations are prone to being discarded during the learning evolution in multimodal models (Zeng et al., 2024; He et al., 2025); and (c) the inaccessibility of cross-modality facial datasets featuring fine-grained attributes. Existing text-face datasets fall into two types: web-scraped low-resolution facial images with inaccurate captions (Li et al., 2024c; Zheng et al., 2022), and close-up facial datasets with limited attributes per caption (only 2 to 7) (Xia et al., 2021; Yu et al., 2023a), which lack detail. Moreover, current facial datasets do not include VQAs, limiting their use for fine-grained understanding tasks. Furthermore, this deficiency directly impacts high-quality face generation (Xiao et al., 2025; Deng et al., 2025; Wang et al., 2025a).

To handle these challenges, we propose ***UniF²ace*** (see Fig. 1), the first UMM specifically tailored for *unified* and *fine-grained* face understanding and generation. UniF²ace aims to address the aforementioned critical challenges by simultaneously performing both tasks and capturing fine-grained facial attributes within a single model. Specifically, we firstly introduce a Dual Discrete Diffusion (D3Diff) loss within a novel theoretical proof that optimizes the negative log-likelihood, significantly improving generation quality. After that, we propose an integrated token-level and sequence-level Mixture-of-Experts(MoE) architecture that adaptively handling semantic and identity facial embeddings, effectively addressing the attribute forgotten phenomenon in representation evolvement and specialized fine-grained representation learning for both understanding and generation tasks. Finally, recognizing the critical role of data, we construct UniF²aceD-1M, a large-scale, specialized dataset containing *130K* facial image-text pairs and *1M* visual question-answering

(VQA) pairs, with *17.7* attributes per caption. Extensive experiments on **UniF²aceD-1M** and other benchmarks demonstrate that UniF²ace significantly outperforms various top-leading single-task models or UMMs with a similar scale and dedicated face models across both understanding and generation tasks, with 7.1% higher Desc-GPT and 6.6% higher VQA-score. Besides, our method achieves comparable or even better accuracy than larger-scale models and establishes a strong baseline. Our main contributions are as follows:

- A unified face understanding and generation framework: We introduce UniF²ace, the first unified multimodal model for fine-grained face understanding and generation, establishing a solid baseline.

- A novel Dual Discrete Diffusion (D3Diff) loss function and a hybrid MoE architecture: We introduce D3Diff, a novel loss function within that theoretically unifies score-based diffusion and masked generative models, leading to a better approximation of the negative log-likelihood for high-fidelity generation and fine-grained attribute control. Additionally, we explore a hybrid Mixture-of-Experts (MoE) architecture at the token and sequence levels, adaptively incorporating the semantic and identity facial embeddings to complement the attribute forgotten phenomenon in representation evolvement.

- We construct UniF²aceD-1M, a dataset containing 1M VQAs with an automated pipeline. Extensive experiments demonstrate that UniF²ace significantly outperforms or is on par with existing state-of-the-art models with a similar scale on various benchmarks, all while providing a more unified and efficient solution.

## 2 RELATED WORK

The field of Unified Multimodal Models (UMMs) has seen significant progress in integrating diverse understanding and generation tasks within single frameworks for generic domains (Ma et al., 2024; Team, 2024; Xie et al., 2024). However, their application to fine-grained visual analysis, especially in the complex domain of human faces, remains largely unexplored. Within the face domain, existing research is primarily fragmented into separate understanding models (often MLLM-based) (Sun et al., 2024a; Xing et al., 2024) and generation models (typically diffusion-based) (Dai et al., 2025; Wang et al., 2024b). Crucially, these approaches often struggle with fine-grained attribute processing and fail to unify understanding and generation effectively. This dual deficiency represents a significant gap that UniF²ace addresses. We also provide a more comprehensive review of Unified Multimodal Models and Face Multimodal Models in **Appendix A**.

## 3 METHODOLOGY

We introduce our unified multimodal model, UniF²ace, designed to model both the understanding and generation of fine-grained facial attributes. Our approach is realized from two perspectives: loss function (Sec. 3.1) and network architecture (Sec. 3.2). Regarding the generation strategy, we recognize that the generation of fine-grained facial attributes is significantly more challenging than understanding tasks, as highlighted in prior studies (Du et al., 2017; Zhou et al., 2024; Xie et al., 2024). To address this, we harness the theory of score matching in discrete diffusion (Lou et al., 2024) and propose the dual discrete diffusion (D3Diff) training strategy, ensuring the meticulous synthesis of facial details. For network architecture, existing UMMs typically focus on dense architectures (Zhou et al., 2024; Xie et al., 2024) or solely on achieving token-level MoE (Deng et al., 2025), lacking the selective integration of instance features. To overcome these limitations, we introduce token-level and sequence-level MoE layers. Distinct MoE modules are designed for generation and understanding tasks, selectively integrating information such as facial embeddings to enhance the model's ability to capture subtle facial attributes.

### 3.1 DUAL DISCRETE DIFFUSION

In generative modeling, masked generative models (Chang et al., 2022) are a widely adopted approach. However, in this section, we introduce discrete score matching and theoretically prove that it offers a better approximation to the negative log-likelihood. We also establish a theoretical connection between the two approaches and finally propose a new loss function to ensure stable optimization, thereby improving the alignment between the generated faces and fine-grained attributes in prompts.

In a discrete diffusion process, each image $\mathbf{x}_0$ is associated with a probability $q(\mathbf{x}_0)$, and its latent distribution at time $t$ under noise adding is denoted by $q(\mathbf{x}_t)$. The forward diffusion is modeled as a continuous-time Markov chain, governed by the linear ordinary differential equation (ODE):

$$\frac{d}{dt}q_{t|s}(\boldsymbol{y} \mid \boldsymbol{x}) = q_{t|s}(\boldsymbol{y} \mid \boldsymbol{x})\,\boldsymbol{Q}_t, \qquad (1)$$

which converges to a stationary distribution as $t \to \infty$. Here, $\boldsymbol{Q}_t$ denotes a time-dependent sequence of transition matrices. The closed-form solution of this ODE can be expressed as $\boldsymbol{Q}_{t|s} = \exp\big((\bar{\sigma}(t) - \bar{\sigma}(s))\,\boldsymbol{Q}\big)$, where $\bar{\sigma}(t) = \int_0^t \sigma(s)\,ds$ represents the cumulative noise level and $\exp$ is the matrix exponential. The reverse process is given by Lou et al. (2024):

$$\frac{dq_{T-t}}{dt} = \tilde{\boldsymbol{Q}}_{T-t}\,q_{T-t}, \quad \tilde{\boldsymbol{Q}} = \frac{q_t(\boldsymbol{y})}{q_t(\boldsymbol{x})}\,\boldsymbol{Q}_t(\boldsymbol{x}, \boldsymbol{y}), \qquad (2)$$

where $\tilde{\boldsymbol{Q}}$ is the reverse diffusion matrix. In our work, we focus on the absorbing state, which is widely used in masked generative models (Chang et al., 2022; Xie et al., 2024). Assuming independence among tokens, as supported by Sahoo et al. (2024); Shi et al. (2025), the exact formulation is deferred to Appendix D. The score-based discrete diffusion model (Lou et al., 2024) introduces a training-stable loss $\mathcal{L}_{\text{score}}(s_\theta)$ to estimate the denoising score. It is defined as:

$$\mathcal{L}_{\text{score}}(s_\theta) = \mathbb{E}_{\mathbf{x}\sim p}\left[\sum_{\mathbf{y}\neq\mathbf{x}} w_{xy}\left(s_\theta(\mathbf{x})_y - \frac{q(\mathbf{y})}{q(\mathbf{x})}\log s_\theta(\mathbf{x})_{\mathbf{y}} + K\left(\frac{q(\mathbf{y})}{q(\mathbf{x})}\right)\right)\right], \qquad (3)$$

where $s_\theta(\mathbf{x}_t, t) \approx \left[\frac{q_t(\boldsymbol{y}_t)}{q_t(\boldsymbol{x}_t)}\right]_{\boldsymbol{y}_t\in\mathcal{X}}$ is the predicted score from the neural network, and $K(a) = a(\log a - 1)$ is a normalizing constant ensuring $\mathcal{L}_{\text{score}} \geq 0$.

To illustrate the advantage of $\mathcal{L}_{\text{score}}$, we start from the negative log-likelihood (NLL), which serves as a fundamental criterion for evaluating the quality of training in generative models. Since exact computation of the NLL is generally infeasible, prior works have derived two different surrogate formulations that upper bound the NLL while remaining computationally tractable. Specifically, one is $\mathcal{L}_1 = \mathcal{L}_{\text{score}}(s_\theta) + D_{\text{KL}}\big(q_{T|0}(\cdot|\mathbf{x}_0) \,\|\, p_{\text{base}}\big)$, which predicts the score (Lou et al., 2024). And the other is $\mathcal{L}_2 = -\sum_{t=1}^T \mathbb{E}_{q(\mathbf{x}_0)\,q(\mathbf{x}_t|\mathbf{x}_0)}\big[\log p_\theta(\mathbf{x}_0|\mathbf{x}_t)\big] - C$, widely used to predict masked tokens (Xie et al., 2024). where $C$ is a residual constant independent of the model parameters (see Appendix E for details). The following theorem formally establishes the relationship between these two surrogates and demonstrates that $\mathcal{L}_1$, which incorporates the score loss, yields a tighter upper bound on NLL.

**Theorem 1.** *Let* $-\log p_\theta(\mathbf{x}_0)$ *denote the negative log-likelihood of the original data distribution. Then the following inequality holds:*

$$-\log p_\theta(\mathbf{x}_0) \leq \mathcal{L}_1 \leq \mathcal{L}_2. \qquad (4)$$

The proof is deferred to Appendix E. Importantly, this result implies that $\mathcal{L}_{\text{score}}$ provides a tighter relaxation of the maximum likelihood objective compared to the masked generative related loss $\mathcal{L}_2$, thereby offering a more precise approximation of the NLL. In practice, the marginal distribution $q(\mathbf{y})$ is often intractable, and the exact analytical form of $q(\mathbf{x})$ is unknown. A key insight is that, in masked generative models, the posterior probability model $p_\theta(\mathbf{x}_0|\mathbf{x}_t)$ can be related to the discrete diffusion score function via Bayes' theorem:

$$p_\theta(\mathbf{x}_0 \mid \mathbf{x}_t) \approx q_t(\mathbf{x}_t|\mathbf{x}_0)\left[\frac{q_t(\mathbf{x}_0)}{q_t(\mathbf{x}_t)}\right]_\theta = q_t(\mathbf{x}_t \mid \mathbf{x}_0)s_\theta(\mathbf{x}_t). \qquad (5)$$

Leveraging this relation, we propose the dual discrete diffusion (D3Diff) loss for training posterior networks:

$$\mathcal{L}_{\text{D3Diff}} = -\sum_{t=1}^T \mathbb{E}_{q(\mathbf{x}_0)q(\mathbf{x}_t|\mathbf{x}_0)}\left[\log p_\theta(\mathbf{x}_0|\mathbf{x}_t)\right] + \alpha\mathcal{L}_{\text{score}}\left(p_\theta(\mathbf{x}_0|\mathbf{x}_t)/q_t(\mathbf{x}_t|\mathbf{x}_0)\right), \qquad (6)$$

where $q(\mathbf{x}_0)$ is the data distribution, $q(\mathbf{x}_t|\mathbf{x}_0)$ is the forward diffusion distribution, and $p_\theta(\mathbf{x}_0|\mathbf{x}_t)$ is the learned posterior parameterized by $\theta$. The score loss $\mathcal{L}_{\text{score}}$ is weighted by a hyperparameter $\alpha$. Eq. 6 establishes a computationally tractable connection between masked generative models and score-based models. Unlike traditional masked generative losses, which rely solely on likelihood, our *D3Diff* loss jointly optimizes two distinct upper bounds of the maximum likelihood objective, enabling stable optimization and fine-grained generation (See Tab. 7).

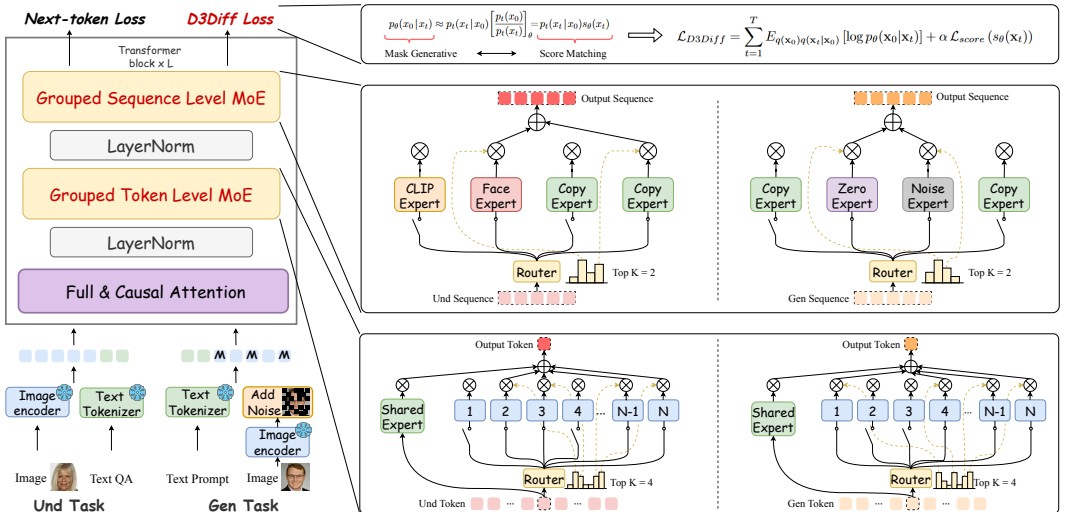

Figure 2: Our UniF²ace centered on two key innovations. First, we design the Transformer with Mixture-of-Experts (MoE) hierarchy: a token-level MoE provides task-specific routing for individual tokens, while a sequence-level MoE injects holistic, domain-specific features. Second, the model's generative capability is optimized by our proposed D3Diff loss, which unifies masked generation with score matching to ensure high-fidelity synthesis of fine-grained facial details.

## 3.2 MULTI-LEVEL GROUPED MIXTURE-OF-EXPERT

To capture fine-grained facial attributes while maintaining facial embeddings, we design distinct MoE layers, termed Multi-level Grouped MoE, tailored for both generation and understanding subtasks. This ensures optimal performance for each task, as illustrated in Fig. 2. We incorporate a sequence-level MoE layer after the token-level MoE layer to effectively process instance-level inputs, such as images and facial embeddings.

**Token-Level MoE.** We partition a feedforward neural network (FFN) into multiple experts with reduced hidden dimensions and use a Top-K activation strategy (Fig. 2). We also employ integrate generalized knowledge across contexts. Unlike prior methods, we introduce grouped MoE, dividing experts into two groups based on the different tasks of Text-to-Image (T2I) and Multimodal Understanding (MMU). Each group combines shared and routed MoE, with expert-level balance loss computed independently per group:

$$\mathcal{L}_{\text{Balance}} = \lambda_{\text{t2i}} \sum_{i=1}^{N_{\text{t2i}}} f_i P_i + \lambda_{\text{mmu}} \sum_{j=1}^{N_{\text{mmu}}} f_j P_j, \qquad (7)$$

where $\lambda_{\text{t2i}}$ and $\lambda_{\text{mmu}}$ are balance factors; $N_{\text{t2i}}$ and $N_{\text{mmu}}$ means the number of routed experts for T2I and MMU tasks, respectively; $f_i$ and $P_j$ denote expert selection frequency and probability.

**Sequence-Level MoE.** We propose sequence-level MoE, where distinct experts process the entire image feature. We design three experts for the T2I group: copy expert (skip operation), zero expert (discard operation), and noise expert. The copy and zero experts require no additional parameters.

Figure 3: Clip/Face Expert enhances the model's understanding of fine-grained facial attributes by incorporating semantic and identity features.

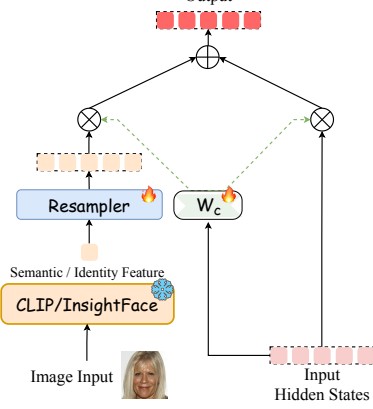

$$\boldsymbol{E}_{\text{copy}}(\boldsymbol{x}) = \mathbf{x} \quad \text{and} \quad \boldsymbol{E}_{\text{zero}}(\boldsymbol{x}) = \mathbf{0}, \qquad (8)$$

where $\boldsymbol{E}_{\text{copy}}(\cdot)$ is the copy expert and $\boldsymbol{E}_{\text{zero}}(\cdot)$ is the zero expert. For the noise expert $\boldsymbol{E}_{\text{noise}}(\cdot)$, we first integrate the time-step embedding, which operates on the noise level $\bar{\sigma}(t)$ to obtain the noise

embedding vector $\mathbf{v}_{\text{noise}}$, following score-based discrete diffusion models (Lou et al., 2023). Then, a resampler $\mathcal{S} : \mathbb{R}^h \to \mathbb{R}^{L \times D}$ maps $\boldsymbol{v}_{\text{noise}}$ into the sequence feature space (see **Appendix E** for resampler details). The resampled noise embedding is added as a matrix to the sequence feature. Formally, the noise expert's output is:

$$\boldsymbol{E}_{\text{noise}}(\mathbf{x}) = w\mathbf{x} + (1 - w)\mathcal{S}(\boldsymbol{v}_{\text{noise}}), \tag{9}$$

$$w = \text{Softmax}(\boldsymbol{W}_{\text{noise}} \cdot \text{Flatten}(\mathbf{x})), \tag{10}$$

where $\boldsymbol{W}_{\text{noise}} \in \mathbb{R}^{2 \times (L \cdot D)}$ is a trainable weight matrix. In the MMU task, we include copy experts and introduce CLIP experts and face experts (See Figure 3), which are similar to noise experts. Next we extract image embeddings by CLIP (Radford et al., 2021) and face embeddings using AntelopeV2 as supplementary features to enhance fine-grained facial attribute capture. Formally, the outputs of the CLIP and face experts are:

$$\boldsymbol{E}_{\text{CLIP}}(\mathbf{x}) = w_{clip}\mathbf{x} + (1 - w_{clip})\mathcal{S}(\mathcal{G}(\boldsymbol{X})), \tag{11}$$

$$\boldsymbol{E}_{\text{face}}(\mathbf{x}) = w_{face}\mathbf{x} + (1 - w_{face})\mathcal{S}(\mathcal{F}(\boldsymbol{X})), \tag{12}$$

where $\mathcal{G}$ and $\mathcal{F}$ are the image encoder and face encoder, respectively. $\boldsymbol{X}$ is the input face image.

### 3.3 OVERALL TRAINING OBJECTIVES

To perform both auto-regressive and discrete score-based diffusion modeling, we employ two learning objectives: 1) Next Token Prediction (NTP) and 2) Dual Discrete Diffusion. Given a sequence with $N$ image tokens $\mathcal{X} = \{\boldsymbol{X}_1, \boldsymbol{X}_2, \dots, \boldsymbol{X}_N\}$ and $M$ text tokens $\mathcal{Y} = \{\boldsymbol{Y}_1, \boldsymbol{Y}_2, \dots, \boldsymbol{Y}_M\}$. Then we maximize the likelihood of text tokens $\mathcal{Y}$ by employing the standard language modeling objective (NTP loss):

$$\mathcal{L}_{\text{MMU}} = \sum_{i=1}^{M} \log P(\boldsymbol{Y}_i \mid \boldsymbol{Y}_{<i}, \mathcal{X}), \tag{13}$$

Next, the overall training objectives of UniF$^2$ace are formulated as:

$$\mathcal{L}_{\text{total}} = \mathcal{L}_{\text{MMU}} + \mathcal{L}_{\text{D3Diff}}, \tag{14}$$

## 4 EXPERIMENT

### 4.1 UNIF$^2$ACED-1M DATASET FOR FINE-GRAINED FACE UNDERSTANDING AND GENERATION

| Dataset | Face Resolution | VQA Availability | Image | Caption Tokens (Avg.) | Face Attributes (Avg. per Caption) |
|---|---|---|---|---|---|
| LAION-Face (Zheng et al., 2022) | Low | ✗ | 50M | 16 | 2.7 |
| FLIP-80M (Li et al., 2024c) | Low | ✗ | 80M | 22 | 4.4 |
| FFHQ-Text (Zhou & Shimada, 2021) | High | ✗ | 760 | 45 | 12.2 |
| MM-CelebA-HQ (Karras et al., 2018) | High | ✗ | 30K | 26 | 6.2 |
| CelebV-Text (Yu et al., 2023b) | High | ✗ | 70K | 80 | 4.3 |
| **UniF$^2$aceD-1M** (Ours) | High | ✓(1M) | **130K** | **120** | **17.7** |

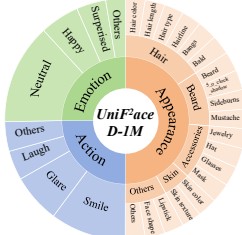

Figure 4: UniF$^2$aceD-1M contains high-resolution facial images, the largest number of facial attributes, 130K fine-grained image-caption pairs and 1 million VQAs.

Existing datasets for multimodal facial modeling frequently suffer from significant limitations, hindering advancements in fine-grained understanding and generation. Common deficiencies include low-resolution imagery, imprecise or web-scraped captions lacking subtle attribute details, and a pervasive absence of comprehensive visual question-answering (VQA) pairs tailored to facial specifics (Li et al., 2024c; Zheng et al., 2022; Xia et al., 2021; Yu et al., 2023a; Karras, 2019). These shortcomings mean that current models struggle to synthesize nuanced facial expressions, comprehend intricate visual semantics, or reason effectively about complex facial attributes. To overcome these challenges and truly enable *unified* and *fine-grained* multimodal facial intelligence, we introduce UniF$^2$aceD-1M. This high-quality dataset serves as a cornerstone of our framework, meticulously designed to bridge these critical data gaps.

As shown in Fig. 4, our UniF$^2$aceD-1M provides a resource distinguished by its fine-grained detail and scale. It comprises nearly 130K high-fidelity facial images, each paired with richly detailed

captions that encompass a wide spectrum of 46 attributes related to appearance, actions, and emotions. This meticulous level of detail is paramount for both robust model training and the generation of highly controllable and realistic facial outputs. Furthermore, a key innovation of UniF$^2$aceD-1M is the inclusion of approximately 1M specialized VQA pairs. Unlike general VQAs, ours are meticulously crafted to probe diverse facial appearances, emotions, and provide detailed reasoning for character actions. This unique VQA collection is specifically designed to enhance MLLMs ability to understand and reason about fine-grained facial attributes through instruction tuning. By offering a substantial collection of high-quality facial imagery, richly detailed captions, and a unique, large-scale set of facial VQAs, UniF$^2$aceD-1M sets a new standard, providing the indispensable resources for developing next-generation unified models capable of sophisticated fine-grained facial intelligence. More collection and operation details can be found in the **Appendix B**.

## 4.2 METRICS AND OTHER FACIAL DATASETS

We rigorously evaluate UniF$^2$ace's performance across both generation and understanding tasks on our UniF$^2$aceD-1M test set. To provide a comprehensive assessment and verify the generalizability of our method, we also conduct evaluations on other public benchmarks, including FFHQ-Text (Zhou & Shimada, 2021), MM-CelebA (Xia et al., 2021), and CelebV-Text (Yu et al., 2023b). **For generation tasks**, we used VQAscore to measure the relevance of generated images to captions, reporting results based on CLIP-FlanT5-11B (VQAscore-CF5) (Lin et al., 2024b) and LLaVA-v1.5-13B (VQAscore-LV) (Liu et al., 2024c) for robust assessment. We also employ Fréchet Inception Distance (FID) to measure similarity to ground truth and VLM-score to evaluate facial realism. **For understanding tasks**, we follow LLaVA (Liu et al., 2023) and use GPT-4o (Hurst et al., 2024) and DeepSeek-v3 (Liu et al., 2024a) to score responses on a 1-10 scale across two dimensions: detailed captioning (Desc-GPT, Desc-DS), assessing accuracy in capturing face attributes, and VQA (Conv-GPT, Conv-DS), measuring precision in responding to fine-grained queries. To fully validate UniF$^2$ace, we compare it with SOTA models. This includes generative models such as autoregressive LlamaGen (Yu et al., 2023c) and diffusion-based Stable Diffusion 3 (SD3) (Esser et al., 2024), as well as leading unified multimodal models (UMMs) like TokenFlow (Qu et al., 2024) and OmniFlow (Li et al., 2024a). More implementations details are in **Appendix C**.

## 4.3 FACE GENERATION

Table 1: Comparison of face generation of UniF$^2$ace with generative-only and UMMs. **Bold** indicates the best, while underlined denotes the best. We use red to highlight the larger-scale model.

| Type | Model | Method | # Params | VQAscore-CF5↑ | VQAscore-LV↑ | FID↓ | VLM-score↑ |
|------|-------|--------|----------|---------------|--------------|------|------------|
| Gen. Only | LlamaGen (Sun et al., 2024b) | AR | 0.8B | 0.746 | 0.551 | 183.466 | 49.773 |
| | DALL-E 3 (Betker et al., 2023) | AR | - | 0.845 | 0.644 | 106.477 | 50.122 |
| | SD3 (Esser et al., 2024) | Diff | 2B | **0.903** | 0.671 | 93.471 | 75.944 |
| | SDXL (Podell et al., 2023) | Diff | 2.6B | 0.876 | 0.660 | 123.095 | 72.764 |
| | Flux.1-dev (Labs, 2024) | Diff | 12B | 0.893 | 0.674 | 76.427 | 84.513 |
| Und. and Gen. | TokenFlow (Qu et al., 2024) | AR | 7B | 0.871 | 0.664 | 98.194 | 73.177 |
| | OmniFlow (Li et al., 2024a) | Diff | 3.4B | 0.798 | 0.585 | 180.933 | 24.960 |
| | JanusFlow (Ma et al., 2024) | AR + Diff | 1.3B | 0.881 | 0.653 | 72.825 | 61.593 |
| | Show-o (Xie et al., 2024) | AR + Diff | 1.3B | 0.855 | 0.650 | 142.557 | 75.618 |
| | UniF$^2$ace(Ours) | AR + Diff | 1.8B | 0.894 | **0.679** | **66.005** | **88.049** |

Table 2: Comparison of face generation on other public datasets. The experimental setup utilized the built-in short captions of datasets as text prompts for generation.

| Type | Model | Params | FFHQ-Text | | | MM-CelebA | | | CelebV-Text | | |
|------|-------|--------|-----------|---|---|-----------|---|---|-------------|---|---|
| | | | VQAScore↑ | FID↓ | VLM-Score↑ | VQAScore↑ | FID↓ | VLM-Score↑ | VQAScore↑ | FID↓ | VLM-Score↑ |
| Gen.Only | LlamaGen (Sun et al., 2024b) | 0.8B | 0.336 | 201.341 | 46.412 | 0.358 | 187.311 | 48.121 | 0.721 | 289.841 | 46.906 |
| | DALL-E 3 (Betker et al., 2023) | - | 0.385 | 196.132 | 49.131 | 0.413 | 158.795 | 49.130 | 0.792 | 295.131 | 54.359 |
| | SD3 (Esser et al., 2024) | 2B | 0.423 | 156.129 | 74.492 | 0.459 | 105.141 | 80.142 | 0.803 | 239.313 | 74.127 |
| | SDXL (Podell et al., 2023) | 2.6B | 0.396 | 181.261 | 64.255 | 0.420 | 139.028 | 73.149 | 0.788 | 271.319 | 70.991 |
| | Flux.1-dev (Labs, 2024) | 12B | 0.434 | 136.360 | 83.621 | 0.467 | 128.462 | **87.764** | **0.806** | 254.043 | 84.901 |
| Gen.&Und. | TokenFlow (Qu et al., 2024) | 7B | 0.409 | 160.023 | 74.349 | 0.421 | 129.562 | 71.092 | 0.781 | 273.972 | 79.526 |
| | OmniFlow (Li et al., 2024a) | 3.4B | 0.376 | 228.094 | 25.431 | 0.368 | 201.413 | 27.892 | 0.800 | 290.131 | 36.839 |
| | JanusFlow (Ma et al., 2024) | 1.3B | 0.413 | 149.231 | 60.984 | 0.445 | 129.131 | 63.418 | 0.797 | 259.236 | 66.587 |
| | Show-o (Xie et al., 2024) | 1.3B | 0.391 | 177.053 | 73.141 | 0.428 | 141.311 | 74.242 | 0.785 | 260.210 | 70.482 |
| | UniF$^2$ace(Ours) | 1.8B | **0.451** | **125.287** | **87.412** | **0.481** | **85.179** | 86.978 | 0.804 | **224.412** | **94.986** |

**Generation Performance on UniF$^2$aceD-1M and Public Dataset.** On our UniFaceD-1M benchmark (Tab. 1), our 1.8B parameter UniF$^2$ace sets a new state-of-the-art, outperforming all competing UMMs on key generation metrics including FID, VQA-score, and VLM-score. Furthermore, the model also demonstrates robust generalization, consistently achieving leading scores on public cross-facial datasets such as FFHQ-Text (Zhou & Shimada, 2021), MM-CelebA (Xia et al., 2021), and CelebV-Text (Yu et al., 2023a) (Tab. 2). This strong and consistent performance validates the effectiveness of

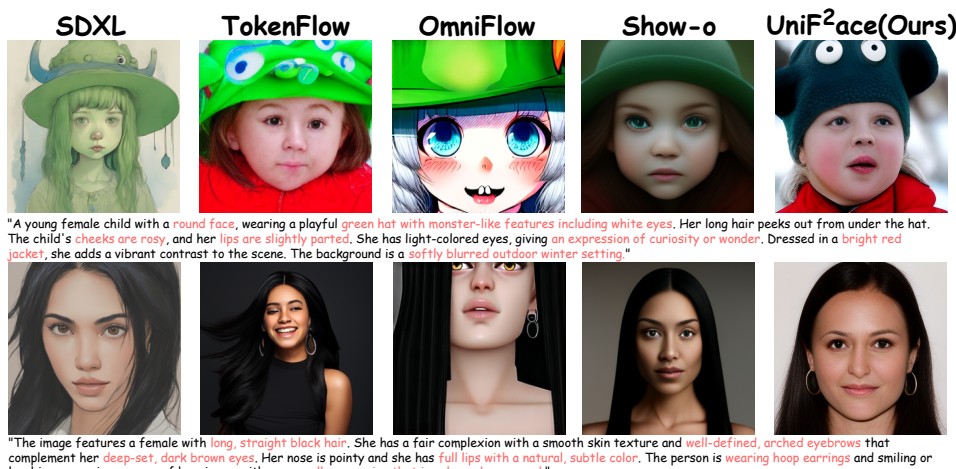

Figure 5: Comparative analysis of face images generation quality across SDXL (Podell et al., 2023), TokenFlow (Qu et al., 2024), OmniFlow (Li et al., 2024a), Show-o (Xie et al., 2024), and UniF$^2$ace. Our proposed UniF$^2$ace effectively captures more detailed information from prompts.

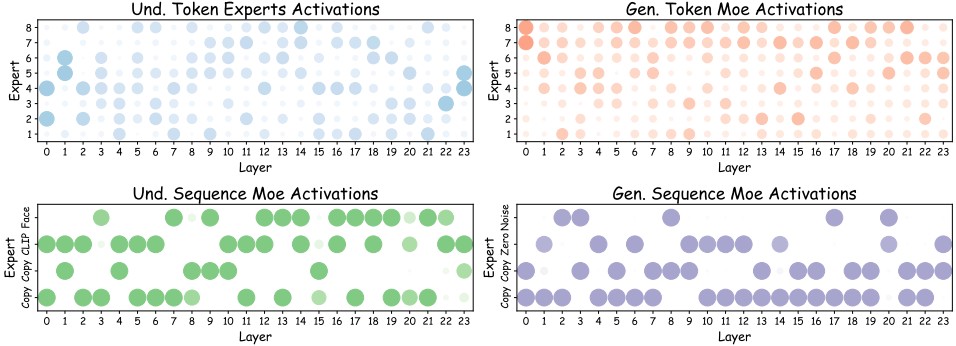

Figure 6: Activation frequency of Token-Level and Sequence-Level MoE in different layers. The left column indicates understanding tasks, while the right column indicates generation tasks. Larger circles indicate experts who are activated more frequently.

our D3Diff loss and multi-level grouped MoE architecture for high-quality, fine-grained facial image generation across diverse settings.

**Visualization Analysis.** As shown in Fig. 5, we conduct qualitative evaluation on challenging UniF$^2$aceD-1M test scenarios involving complex facial details. UniF$^2$ace excels at generating realistic faces that capture fine-grained details from complex prompts (e.g., "rosy cheeks," "hoop earrings"), visibly outperforming other models. More examples can be found in Fig. 8 and Fig. 9. Besides, as shown in Fig. 6, we analyze MoE activation frequencies across layers. For token-level MoEs, high activation frequencies are concentrated between experts 5 and 8, indicating limited token feature variability in the generation task. For sequence-level MoEs, noise and zero expert activations are evenly distributed, indicating effective training with selective noise embedding and truncation.

## 4.4 FACE UNDERSTANDING

Table 3: Comparison of face understanding of UniF$^2$ace with understanding-only and UMMs.

| Type | Model | Method | # Params | Desc-GPT↑ | Conv-GPT↑ | Desc-DS↑ | Conv-DS↑ |
|------|-------|--------|----------|-----------|-----------|----------|----------|
| Und. Only | VILA1.5 (Lin et al., 2023) | AR | 3B | 4.76 | 5.20 | 6.56 | 6.54 |
| | Qwen2-VL (Wang et al., 2024a) | AR | 7B | 5.16 | 6.27 | 5.50 | 6.86 |
| | LLaVA-v1.5 (Liu et al., 2024b) | AR | 7B | 4.28 | 5.48 | 4.84 | 6.20 |
| | InternVL2.5 (Chen et al., 2024b) | AR | 8B | 5.62 | 5.89 | 6.30 | 6.55 |
| | Qwen2.5-VL (Bai et al., 2025) | AR | 3B | 4.88 | 6.38 | 4.98 | 6.75 |
| Und. and Gen. | TokenFlow (Qu et al., 2024) | AR | 7B | 5.02 | 5.80 | 5.82 | 6.39 |
| | OmniFlow (Li et al., 2024a) | Diff | 3.4B | 1.62 | - | 1.90 | - |
| | JanusFlow (Ma et al., 2024) | AR + Diff | 1.3B | 4.88 | 6.06 | 5.42 | 6.77 |
| | Show-o (Xie et al., 2024) | AR + Diff | 1.3B | 3.88 | 4.17 | 5.24 | 4.90 |
| | UniF$^2$ace(Ours) | AR + Diff | 1.8B | **6.02** | **6.53** | **7.38** | **7.29** |

Table 4: Comparison of face understanding on other public datasets. The experiments utilized the dataset's captions as labels for captioning task evaluation, showing the robustness of UniF$^2$ace.

| Type | Model | Params | FFHQ-Text | | MM-CelebA | | CelebV-Text | |
|---|---|---|---|---|---|---|---|---|
| | | | Desc-GPT | Desc-DS | Desc-GPT | Desc-DS | Desc-GPT | Desc-DS |
| Und.Only | VILA1.5 (Lin et al., 2023) | 3B | 4.29 | 4.79 | 4.48 | 4.59 | 4.61 | 4.76 |
| | Qwen2-VL (Wang et al., 2024a) | 7B | 4.68 | 5.41 | 5.11 | 5.40 | 4.90 | 4.95 |
| | LLaVA-v1.5 (Liu et al., 2024b) | 7B | 4.01 | 4.60 | 4.29 | 4.26 | 4.54 | 4.50 |
| | InternVL2.5 (Chen et al., 2024b) | 8B | 5.09 | 5.58 | 4.75 | 4.98 | 5.07 | 5.01 |
| | Qwen2.5-VL (Bai et al., 2025) | 3B | 4.38 | 4.92 | 4.72 | 4.70 | 5.20 | 5.10 |
| Gen.&Und. | TokenFlow (Qu et al., 2024) | 7B | 5.04 | 5.75 | 4.99 | 5.01 | 4.86 | 5.10 |
| | OmniFlow (Li et al., 2024a) | 3.4B | 2.83 | 3.06 | 3.41 | 3.38 | 2.90 | 3.03 |
| | JanusFlow (Ma et al., 2024) | 1.3B | 4.31 | 5.15 | 4.60 | 4.71 | 4.54 | 4.86 |
| | Show-o (Xie et al., 2024) | 1.3B | 3.86 | 4.67 | 4.38 | 4.39 | 4.49 | 4.57 |
| | UniF$^2$ace (Ours) | 1.8B | **5.12** | **5.92** | **6.24** | **6.80** | **5.87** | **5.29** |

Table 5: Performance comparison of understanding on FaceXBench (Narayan et al., 2025).

Table 6: Performance comparison of non-celebrity face understanding.

| Model | Params | Attr. Exp. | Bias Fair. | Face Rec. | Face Loc. | Deepfake |
|---|---|---|---|---|---|---|
| Qwen2-VL (Wang et al., 2024a) | 7B | 31.25 | 35.11 | **38.00** | 39.30 | 71.33 |
| Qwen2.5-VL (Bai et al., 2025) | 3B | 37.50 | 33.78 | 37.33 | 32.40 | 71.33 |
| UniF$^2$ace (Ours) | 1.8B | **39.50** | **37.56** | 37.33 | **39.60** | **72.67** |

| Model | Params | Desc-GPT ↑ | Desc-DS ↑ |
|---|---|---|---|
| Qwen2-VL (Wang et al., 2024a) | 7B | 4.97 | 5.13 |
| Qwen2.5-VL (Bai et al., 2025) | 3B | 5.30 | 5.51 |
| UniF$^2$ace (Ours) | 1.8B | **5.54** | **5.77** |

**Understanding Performance on UniF$^2$aceD-1M and Public Dataset.** On our UniF$^2$aceD-1M benchmark (Tab. 3), UniF$^2$ace sets a new state-of-the-art in fine-grained facial understanding, achieving the highest scores across all metrics. Crucially, it surpasses even larger, specialized models like InternVL2.5 (8B) and all competing UMMs. Furthermore, as shown in Tab. 4, this superior understanding capability demonstrates strong generalization, as UniF$^2$ace also consistently achieves top captioning scores on public cross-facial datasets. This robust performance across diverse benchmarks validates the effectiveness of our approach in learning transferable, fine-grained facial representations, affirming UniF$^2$ace's leading position in comprehensive multimodal facial understanding.

**Evaluation on FaceXBench.** We further evaluated our model on FaceXBench (Narayan et al., 2025), a comprehensive benchmark designed for assessing Multimodal Large Language Models (MLLMs). Since the original FaceXBench includes multi-image reasoning tasks that exceed the current single-image inference capability of UniF$^2$ace, we curated a specialized evaluation subset by extracting all single-image test instances. This robust subset comprises a total of 2,150 instances, specifically covering Attribute Expression (400), Bias Fairness (450), Face Recognition (150), Face Localization (1,000), and Face Anti-Spoofing (FAS) & Deepfake (150). Given its significant scale and task diversity, as shown in Table 5, this refined evaluation set provides a rigorous and comprehensive assessment of single-image facial understanding capabilities.

**Evaluation on non-celebrity inputs.** We benchmarked our model against Qwen2-VL(7B) (Wang et al., 2024a) and Qwen2.5-VL(3B) (Bai et al., 2025) using 3,000 random samples from FLIP-80M (Li et al., 2024c) to assess robustness in face understanding. Distinguished from standard celebrity-centric datasets, FLIP-80M captures the complexity of real-world data, including web-scraped and AI-generated content. The dataset presents substantial challenges regarding inconsistent lighting, blur, and unconstrained compositions, consisting largely of low-resolution, non-celebrity images. The results are shown in Table 6, demonstrating the robustness of UniF$^2$ace.

**Quantitative Evaluation.** We compare UniF$^2$ace with advanced autoregressive multimodal models like Qwen2-VL (Wang et al., 2024a) (7B) and similar UMMs (Wu et al., 2024c; Xie et al., 2024). Visual comparisons (right Fig.) confirm UniF$^2$ace's superior fine-grained understanding in VQA tasks, where it accurately identifies subtle details like "sideburns" and "earrings" that competitors miss. We also provide more examples for captioning in the Appendix (**Fig. 10**). Besides, as shown in Fig. 6 (left column), we analyze MoE activation frequencies in the understanding task. Token-level MoEs often select the same expert in the top (the closest to the prediction head) and bottom layers. For sequence-level MoEs, face and CLIP experts are more frequently activated in layers closer to the

Q: Can you describe his hairstyle?

**VILA1.5:** The man in the image has a messy, unkempt hairstyle with his hair sticking up in all directions. score:4

**JanusFlow:** The person has a messy, spiky hairstyle with a lot of volume and texture. The hair is light brown and appears to be somewhat unkempt. score:8

**Uni$^2$Face:** Certainly! He has brown hair that is thick and wavy, styled in a slightly tousled manner. His hairstyle also includes sideburns that complement the overall look. score:9

Q: Is she wearing any earrings?

**InternVL2.5:** The image is not clear enough to determine if she is wearing any earrings. score:3

**Show-o:** Yes, the woman is wearing earrings. score:6

**Uni$^2$Face:** Yes, she is wearing earrings, which add to her stylish appearance. score:7

top, indicating that deeper layers benefit from visual embeddings to better understand face images. Notably, activation patterns are different across the generation groups, highlighting the effectiveness of our group-based strategy.

### 4.5 ABLATION STUDIES

**Coefficient in Dual Discrete Diffusion.** As shown in Table 7, we analyze the D3Diff loss. The optimal weight is $\alpha = 0.01$, which balances the $200\times$ magnitude difference between the score-matching and masked generative losses. The complete D3Diff loss significantly outperforms using either loss component individually. Crucially, the superiority of the score-only loss over the masked-only loss empirically validates our theoretical analysis in **Appendix E**.

Table 7: Ablation study with different loss weights.

| Loss Type | Weight $\alpha$ | VQAscore-CF5↑ | VQAscore-LV↑ | FID↓ | VLM-score↑ |
|---|---|---|---|---|---|
| Only Mask | 0 | 0.879 | 0.661 | 77.463 | 85.993 |
| Only Score | 0.01 | 0.886 | 0.670 | 69.694 | 87.951 |
| D3Diff | 0.1 | 0.887 | 0.673 | 68.903 | 86.378 |
| | 0.01 | **0.894** | **0.679** | **66.005** | 88.049 |
| | 0.001 | 0.884 | 0.668 | 72.736 | **89.220** |

Table 8: Ablation of Face and CLIP Expert.

| Expert Type | | Understanding | | | |
|---|---|---|---|---|---|
| Face | CLIP | Desc-GPT↑ | Conv-GPT↑ | Desc-DS↑ | Conv-DS↑ |
| ✗ | ✗ | 5.21 | 5.31 | 6.27 | 6.36 |
| ✓ | ✗ | 5.67 | 5.93 | 6.86 | 7.10 |
| ✗ | ✓ | 5.81 | 5.46 | 7.12 | 5.84 |
| ✓ | ✓ | **6.02** | **6.53** | **7.38** | **7.29** |

Table 9: Ablation of Top-k in Seq-level MoE.

| Top-K | Generation | | | Understanding | |
|---|---|---|---|---|---|
| | VQAscore↑ | FID↓ | VLM-score↑ | Desc↑ | Conv↑ |
| 1 | 0.879 | 74.914 | 74.314 | 6.57 | 6.42 |
| 2 | 0.894 | 66.005 | **88.049** | **7.38** | 7.29 |
| 3 | 0.895 | 65.413 | 85.401 | 7.26 | 7.34 |
| 4 | **0.897** | **63.632** | 87.795 | 7.23 | **7.36** |

**Ablation of MoEs Architecture.** We ablate our multi-level MoE design ( Tab. 8, Tab. 9, Tab. 10). The results (Tab. 10) first confirm that combining token-level and sequence-level MoEs achieves the best performance, with each component individually outperforming the non-MoE baseline. We further analyze the sequence-level MoE, finding that: (1) for understanding tasks, using both CLIP and Face experts is optimal in fine-grained facial understanding (Tab. 8); and (2) a Top-k=2 selection strategy provides the best balance between performance and efficiency (Tab. 9). These findings validate our hierarchical and specialized MoE design.

Table 10: Ablation study of token- and sequence-level MoE.

| Token MoE | Sequence MoE | Generation | | | Understanding | |
|---|---|---|---|---|---|---|
| | | VQAscore↑ | FID↓ | VLM-score↑ | Desc↑ | Conv↑ |
| ✗ | ✗ | 0.878 | 72.877 | 84.432 | 4.988 | 6.031 |
| ✓ | ✗ | 0.887 | 67.415 | 87.917 | 5.678 | 6.495 |
| ✗ | ✓ | 0.889 | 69.312 | 86.790 | 5.864 | 6.247 |
| ✓ | ✓ | **0.894** | **66.005** | **88.049** | **6.023** | **6.532** |

## 5 CONCLUSION

This paper introduces UniF$^2$ace, the first unified multimodal model (UMM) designed for fine-grained face understanding and generation. The model bridges the gap between score-based models and masked generative models in discrete diffusion, while leveraging token-level and sequence-level mixture-of-experts (MoE) to sparsify the model. Extensive experiments show that UniF$^2$ace outperforms existing UMMs and even surpasses larger generation-only or understanding-only models. This underscores the potential of our improvements to guide future research in face applications of UMM. Additionally, we constructed a large-scale face-text aligned dataset, UniF$^2$aceD-1M, to further advance multimodal research in the community.

## 6 ACKNOWLEDGMENTS

This work was supported by the National Natural Science Foundation of China (Grant Nos. 62502317, 92570101, and 62320106007) and the Science and Technology Development Special Fund Project of Xinjiang Production and Construction Corps (Grant No. 2025AB027). Partial support was also provided by the NUS Start-up Grant (A-0010106-00-00) and the Zhongguancun Academy (Grant No. 1800302002).

## 7 ETHICS STATEMENT

I read all respects with the ICLR Code of Ethics `https://iclr.cc/public/CodeOfEthics` and the research conducted in the paper complies in all respects.

## 8 REPRODUCIBILITY STATEMENT

This paper fully discloses all the source code needed to reproduce the main experimental results in the supplementary material. Besides, we also provide the a complete description of the proposed dataset for their data processing steps in the Appendix B. Finally, we also provide clear explanations of our assumptions, and a complete proof of the claims can be included in the Appendix D, E, and F.

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

# A    RELATED WORKS

**Unified Multimodal Models.**    Recent works (Ma et al., 2024; Li et al., 2024d; Wu et al., 2024a; Chen et al., 2025c; Wang et al., 2024c; Fang et al., 2022; 2026a;b; Tong et al., 2025; Xu et al., 2024; Zeng et al., 2025) in image understanding and generation have primarily focused on unified multimodal models (UMMs). Early approaches (Li et al., 2024b; Wu et al., 2024b) often integrated external decoders of diffusion models (DMs) with text autoregressive models (ARMs). Inspired by next-token prediction tasks, they proposed using a single Transformer (Vaswani et al., 2017) model to unify understanding and generation (Wu et al., 2024c). For instance, Janus-Pro (Chen et al., 2025c) decouples the visual encoder into specialized tokenizers for separate handling of understanding and generation tasks. Chameleon (Team, 2024) and Emu3 (Wang et al., 2024c) employ an ARM to simultaneously manage both tasks, highlighting the advantages of autoregressive models in multitask settings. Additionally, Transfusion (Zhou et al., 2024) and Show-o (Xie et al., 2024) combine a text ARM with a visual DM, enabling seamless integration of image understanding and generation. These studies have advanced the fusion of visual and text generation models, enhancing performance on multimodal tasks. However, despite the proliferation of UMMs, their application has largely been limited to generic domain tasks, with limited exploration in fine-grained visual analysis, particularly in the face domain. Unlike previous UMMs that simply combine ARMs and DMs, we pioneer sparse UMMs by introducing both token-level and sequence-level Mixture of Experts (MoEs), significantly improving model performance.

**Face Multimodal Models.**    Face multimodal models are primarily categorized into two types: face understanding models and face generation models. For understanding, early models were task-specific and lacked multimodality (Miyato et al., 2018; Zhang et al., 2024; Wang et al., 2023; Lee et al., 2023). Recent works (Chettaoui et al., 2025; Sun et al., 2024a; Xing et al., 2024; Zhao et al., 2024) leverage the reasoning capabilities of LLMs or MLLMs, often using MLLM-generated face Q&A data to fine-tune or post-train foundation models, incorporating face domain knowledge. For example, EMO-LLaMA (Xing et al., 2024) introduces facial experts to extract facial features, which are aggregated with handcrafted prompts and fed into LLaMA (Touvron et al., 2023), enabling it to answer facial-related queries. Recent research (Wang et al., 2025b; Li et al., 2025; Zhao et al., 2025) has increasingly focused on performing fine-grained facial attribute analysis. On the modeling front, FaceInsight (Li et al., 2025) advances facial perception by introducing visual-textual alignment of facial knowledge and segmentation maps. FaVChat Zhao et al. (2025) extends these fine-grained perceptual capabilities to the domain of video understanding. Complementing these modeling advancements, a new wave of benchmarks has emerged for rigorous evaluation (Narayan et al., 2025; Wang et al., 2025c). A notable example is FaceXBench (Narayan et al., 2025), which provides a comprehensive assessment covering 14 tasks across 6 broad categories, including bias and fairness, authentication, recognition, and analysis. Collectively, these synergistic efforts in both model development and evaluation are driving the field of fine-grained face understanding forward. For generation, recent works (Dai et al., 2025; Wang et al., 2024b; Huang et al., 2023; Kim et al., 2024) focus on using diffusion models to personalize face images by conditioning on textual and visual information, such as semantic masks, but avoid directly capturing fine-grained face attributes from text prompts. Despite these advances in understanding and generation separately, developing unified multimodal models (UMMs) remains a significant research challenge. Addressing this gap can enhance cross-modal capabilities and advance progress toward Artificial General Intelligence (AGI).

# B    DATASET CONSTRUCTION

To overcome the limitations of existing datasets in the realm of multimodal facial modeling, we introduce a high-quality dataset called *UniF$^2$aceD-1M*, which boasts a remarkable alignment between facial images and textual descriptions (see Fig. 7). This dataset encompasses nearly 130K facial images, each paired with richly detailed captions. Additionally, it contains approximately 1M visual question answers, significantly enhancing its value for training and evaluating multimodal models. By offering such a comprehensive resource, we aim to propel advancements in facial image understanding and generation, establishing a solid foundation for a wide range of multimodal learning tasks. The creation of **UniF$^2$aceD-1M** encompassed three key stages. **(1) Step-1**: Collect high-quality facial

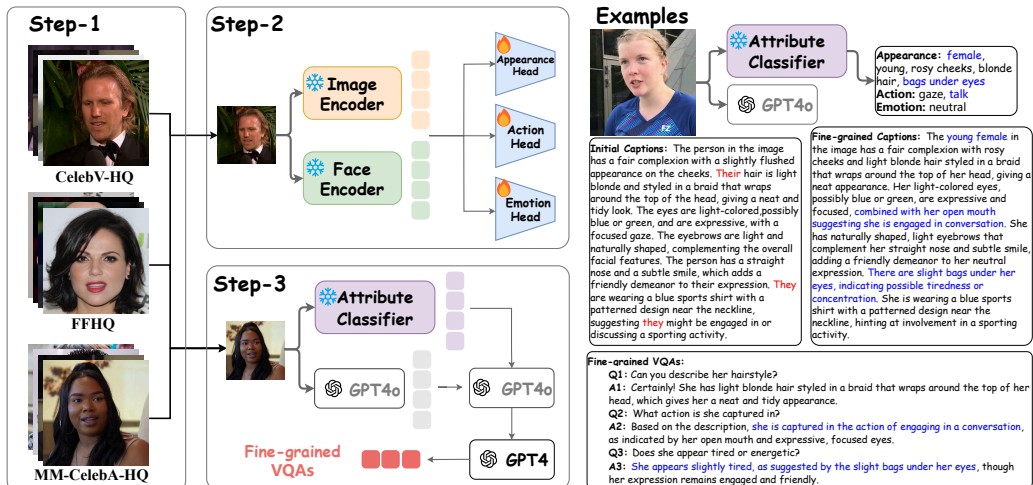

Figure 7: Pipeline and examples of UniF$^2$aceD-1M construction. Left: A three-stage pipeline for building UniF$^2$aceD-1M. Step-1: High-quality face images are collected. Step-2: Detailed captions are generated by GPT-4o with a face attribute model trained to classify fine-grained appearance, action, and emotion. Step-3: Question-answering pairs are created. These stages collectively refine GPT-4o-generated captions and produce fine-grained descriptions for VQAs generation. Right: A representative example showcasing UniF$^2$aceD-1M's ability to correct (*e.g.,* gender), enhance (*e.g.,* bags under eyes), and reason (*e.g.,* talking, slight tiredness) in GPT-4o-generated captions.

images. (2) **Step-2**: Generate detailed captions. (3) **Step-3**: Create question-answering pairs. Each stage is outlined in detail below.

**(1) Step-1: Collect High-quality Facial Images.** In this step, we curated more than 130,000 high-quality facial images from the following distinguished datasets. CelebV-HQ (Zhu et al., 2022) is a large-scale video dataset featuring 35,666 clips representing 15,653 identities, each clip meticulously annotated with 83 facial attributes. We extracted one key frames from each video to utilize detailed annotations for fine-grained face-text alignment. Flickr-Faces-HQ (FFHQ) (Karras, 2019) provided 70,000 high-quality PNG images at a resolution of 1024 by 1024, offering substantial diversity in attributes such as age and ethnicity. Multi-Modal-CelebA-HQ (MM-CelebA-HQ) (Xia et al., 2021) contributed 30,000 high-resolution images paired with descriptive captions that have proven invaluable for facial generation and analysis.

**(2) Step-2: Generate Detailed Captions.** Existing face image datasets often lack detailed descriptions of fine-grained attributes like bags under eyes or jewelry. To handle this, we develop a two-stage caption generation process.

In Stage I, we employed an advanced MLLM such as GPT-4o (Hurst et al., 2024) to produce initial captions. We designed a specialized prompt that incorporated brief face descriptions from the MM-CelebA-HQ dataset (Xia et al., 2021) to help GPT-4o accurately describe key facial attributes, including appearance, emotion, and actions. The detailed descriptions of all prompts are presented later (see Fig. 11).

In Stage II, we refined these captions by training face attribute classification models using the CelebV-HQ dataset (Zhu et al., 2022). Focusing on single-person images, we used the pretrained face model AntelopeV2[1] to extract face embeddings. By combining these with image embeddings from CLIP (Radford et al., 2021), we trained classification heads for appearance, action, and emotion attributes. We selected 29 appearances with accuracies over 93%, 10 actions with accuracies over 87%, and 7 emotions with accuracies over 80% as final preditions for inference. These highly accurate attributes were then predicted for all remaining images in FFHQ and MM-CelebA-HQ datasets (Karras, 2019; Xia et al., 2021). Finally, a prompt integrating these classification results with the Stage I captions was fed into GPT-4o to generate final captions that are both highly accurate and diverse.

---

[1]https://github.com/deepinsight/insightface

**(3) Step-3: Create Question-answering Pairs.** In this step, we proposed 1M VQAs covering diverse facial appearances, emotions, and character action reasoning for our UniF$^2$aceD-1M dataset. These VQAs are designed to enhance MLLMs' ability to understand fine-grained facial attributes through instruction tuning. Inspired by LLaVA (Liu et al., 2024c), we carefully designed prompts to enable GPT-4 (Achiam et al., 2023) to generate a series of VQAs based on image captions, facilitating fine-grained understanding and reasoning. Most current face-text datasets lack VQAs, while VQAs in general image-text datasets often focus on people's clothing, location, and behavior, neglecting detailed facial descriptions. In contrast, our proposed VQAs encompass diverse facial details, including hair, nose, eyes, mouth, ears, skin, eyebrows, and adornments. Additionally, since facial attributes can reflect a character's ongoing actions, our VQAs incorporate detailed reasoning processes to infer and describe these actions. By organizing the VQAs into the same format as the LLaVA dataset (Liu et al., 2024c), we streamlined the process of adapting multimodal face models for post-training. This alignment minimizes alteration costs, ensuring efficient integration and enabling the models to leverage both datasets seamlessly for improved performance.

**Discussion of ethical risks.** Regarding the ethical risks of multimodal models (Gu et al., 2024; 2026), we first verified that the licenses for CelebV-HQ, FFHQ, and MM-CelebA-HQ permit non-commercial research and educational use. Our contribution, which consists of secondary annotations for these datasets, was executed with a focus on neutrality to prevent bias, and every LLM-generated annotation underwent manual review to ensure the absence of hallucinations. To prevent future misuse, the released dataset will explicitly state these terms and strictly inherit the licensing restrictions of its source data.

## C  IMPLEMENTATIONS DETAILS

We train our model on the UniF$^2$aceD-1M training dataset part, comprising 120K $256 \times 256$ face images, each annotated with detailed captions and seven to eight VQAs, about 900K. UniF$^2$ace utilizes discrete image tokens as input, represented by the pre-trained MAGVIT-v2 (Yu et al., 2023c). For token-level MoE, each group (generation and understanding tasks) includes one shared expert and eight routed experts, selected via a top-2 strategy. The expert structure is a single-layer MLP with the gating mechanism (Dai et al., 2024). In sequence-level MoE, the generation group employs two copy experts, one zero expert, and one noise expert. Noise embedding is implemented using sinusoidal embedding, following (Nichol et al., 2021). The noise resampler uses a 4-layer Multi-Head Attention mechanism to map noise embeddings to the UniF$^2$ace hidden space. For the understanding group, there are two copy experts, one CLIP expert, and one face expert. We employ pre-trained self-supervised models (Lin et al., 2025), such as CLIP-ViT for image embedding and AntelopeV2 for face embedding, with a resampler configuration consistent with that of the noise expert. Moreover, training is divided into two stages: Stage I uses only captions for generation and understanding tasks, while Stage II incorporates VQAs into the understanding task. This pipeline transitions the model from general image feature understanding to fine-grained feature capture. Both stages are trained on 8 NVIDIA A100 (80GB) GPUs, optimized using AdamW with a weight decay of 0.01, 5K warm-up steps, and an initial learning rate of 5e-5 with cosine scheduling. The total batch size is 600 for Stage I and 480 for Stage II, with 20K steps for Stage I and 40K steps for Stage II. For a fair comparison, we also performed full-parameter fine-tuning on all competing models using an identical amount of data, leveraging their official fine-tuning scripts where available. In the inference process of UniF$^2$ace, following the computation method in (Lin et al., 2024a), we compute the maximum and minimum activation parameters for UniF$^2$ace under the Top-2 strategy due to the different number of parameters included between different experts in the sequence-level MoE. The total number of parameters for UniF$^2$ace is 1.84B, the maximum activation parameter is about 1.63B, and the minimum activation parameter is about 1.42B. The average number of activation parameters tested in the UniF$^2$aceD-1M test dataset is 1.47B.

## D ABSORBING-STATE CASE WITH INDEPENDENCE BETWEEN TOKENS.

The absorbing-state case means that for any single token $x$ with possible values in $\mathcal{X} = \{1, \ldots, N\}$, the transition matrix is

$$
\boldsymbol{Q}^{\text{absorb}} =
\begin{bmatrix}
-1 & 0 & \cdots & 0 & 1 \\
0 & -1 & \cdots & 0 & 1 \\
\vdots & \vdots & \ddots & \vdots & \vdots \\
0 & 0 & \cdots & -1 & 1 \\
0 & 0 & \cdots & 0 & 0
\end{bmatrix}.
\tag{15}
$$

The reverse transition rate matrix of the reverse process from state $\mathbf{x}_t$ to state $\hat{\mathbf{x}}_t$ is

$$
\tilde{\boldsymbol{Q}}_t \left( \mathbf{x}_t, \hat{\mathbf{x}}_t \right) =
\begin{cases}
\frac{q_t(\hat{\mathbf{x}}_t)}{q_t(\mathbf{x}_t)} \boldsymbol{Q}_t \left( \hat{\mathbf{x}}_t, \mathbf{x}_t \right), & \hat{\mathbf{x}}_t \neq \mathbf{x}_t \\
- \sum_{k \neq \mathbf{x}_t} \tilde{\boldsymbol{Q}}_t \left( \mathbf{x}_t, k \right), & \hat{\mathbf{x}}_t = \mathbf{x}_t
\end{cases}.
\tag{16}
$$

As $\boldsymbol{Q}_t \left( \hat{\mathbf{x}}_t, \mathbf{x}_t \right)$ is known, it is sufficient to estimate the concrete score $\frac{q_t(\hat{\mathbf{x}}_t)}{q_t(\mathbf{x}_t)}$ by a score network $s_\theta \left( \mathbf{x}_t, t \right) \approx \left[ \frac{q_t(\hat{\mathbf{x}}_t)}{q_t(\mathbf{x}_t)} \right]_{\hat{\mathbf{x}}_t \in \mathcal{X}}$. Score based discrete diffusion model is an effective objective to train the score network (Meng et al., 2022; Lou et al., 2023). Specifically, the score function in a multidimensional discrete space is

$$
\boldsymbol{s}_\theta \left( \boldsymbol{x}_t, t \right)_{\hat{\boldsymbol{x}}_t} = s_\theta \left( \mathbf{x}_t^1 \ldots \mathbf{x}_t^i \ldots \mathbf{x}_t^d, t \right) \left[ i, \widehat{\mathbf{x}}_t^i \right] \approx \frac{q_t \left( \mathbf{x}_t^1 \ldots \widehat{x}_t^i \ldots \mathbf{x}_t^d \right)}{q_t \left( \mathbf{x}_t^1 \ldots \mathbf{x}_t^i \ldots \mathbf{x}_t^d \right)},
\tag{17}
$$

and accordingly,

$$
\tilde{\boldsymbol{Q}}_t \left( \mathbf{x}_t^1 \ldots \mathbf{x}_t^i \ldots \mathbf{x}_t^d, \mathbf{x}_t^1 \ldots \widehat{\mathbf{x}}_t^i \ldots \mathbf{x}_t^d \right) \approx \boldsymbol{Q}_t \left( \widehat{\mathbf{x}}_t^i, \mathbf{x}_t^i \right) \boldsymbol{s}_\theta \left( \mathbf{x}_t^1 \ldots \mathbf{x}_t^i \ldots \mathbf{x}_t^d, t \right) \left[ i, \widehat{x}_t^i \right].
\tag{18}
$$

## E PROOF OF THEOREM 1

To prove Theorem 1, we first introduce two loss formulas and establish useful lemmas.

(1) $\mathcal{L}_1 = \mathcal{L}_{\text{score}} \left( s_\theta \right) + D_{KL} \left( q_{T|0} \left( \cdot \mid \mathbf{x}_0 \right) \| q_{\text{base}} \right)$, where $\mathcal{L}_{\text{score}} \left( \mathbf{x}_0 \right)$ is the diffusion weighted denoising score entropy for data point $\mathbf{x}_0$, and $s_\theta = \frac{q_\theta(\mathbf{x}_0|\mathbf{x}_t)}{q(\mathbf{x}_t|\mathbf{x}_0)}$

$$
\begin{aligned}
\mathcal{L}_{\text{score}} \left( s_\theta \right) = \int_0^T \mathbb{E}_{\mathbf{x}_t \sim q_{t|0}(\cdot|\mathbf{x}_0)} \sum_{\mathbf{y} \neq \mathbf{x}_t} & Q_t \left( \mathbf{x}_t, y \right) \left( s_\theta \left( \mathbf{x}_t, t \right)_{\mathbf{y}} \right. \\
& \left. - \frac{q_{t|0} \left( \mathbf{y} \mid \mathbf{x}_0 \right)}{q_{t|0} \left( \mathbf{x}_t \mid \mathbf{x}_0 \right)} \log s_\theta \left( \mathbf{x}_t, t \right)_{\mathbf{y}} + K \left( \frac{q_{t|0} \left( \mathbf{y} \mid \mathbf{x}_0 \right)}{q_{t|0} \left( \mathbf{x}_t \mid \mathbf{x}_0 \right)} \right) \right) dt.
\end{aligned}
\tag{19}
$$

(2) $\mathcal{L}_2 = - \sum_{t=1}^{T} \mathbb{E}_{q(\mathbf{x}_0) q(\mathbf{x}_t | \mathbf{x}_0)} \left[ \log p_\theta \left( \mathbf{x}_0 \mid \mathbf{x}_t \right) \right] - C$, where $C$ is a constant independent of the model parameters. By (Xie et al., 2024), $C = C_1 + C_2$, and constants $C_1$ and $C_2$ are

shown as:

$$
\begin{aligned}
C_1 &= \mathbb{E}_{q(\mathbf{x}_{0:T})}\left[-\sum_{t=1}^{T}\log q\left(\mathbf{x}_t \mid \mathbf{x}_{t-1}\right) + \underbrace{\log p\left(\mathbf{x}_T\right)}_{\text{Note that } p(\mathbf{x}_T)=q(\mathbf{x}_T)}\right] \\
&= \mathbb{E}_{q(\mathbf{x}_{0:T})}\left[-\sum_{t=1}^{T}\log q\left(\mathbf{x}_t,\mathbf{x}_{t-1}\right) + \sum_{t=0}^{T}\log q\left(\mathbf{x}_t\right)\right] \\
C_2 &= \mathbb{E}_{q(\mathbf{x}_{0:T})}\left[\sum_{t=1}^{T}\log q\left(\mathbf{x}_{t-1}\mid\mathbf{x}_t\right)\right] - \mathbb{E}_{q(\mathbf{x}_{0:T})}\left[\sum_{t=1}^{T}\sum_{\tilde{\mathbf{x}}_0} q\left(\tilde{\mathbf{x}}_0\mid\mathbf{x}_{t-1}\right)\log q\left(\tilde{\mathbf{x}}_0\mid\mathbf{x}_t\right)\right] \\
&= \mathbb{E}_{q(\mathbf{x}_{0:T})}\left[\sum_{t=1}^{T}\log q\left(\mathbf{x}_t,\mathbf{x}_{t-1}\right) - \sum_{t=1}^{T}\log q\left(\mathbf{x}_t\right)\right] \\
&\quad - \sum_{t=1}^{T}\mathbb{E}_{q(\mathbf{x}_{0:T})q(\tilde{\mathbf{x}}_0\mid\mathbf{x}_{t-1})}\left[\log q\left(\tilde{\mathbf{x}}_0\mid\mathbf{x}_t\right)\right].
\end{aligned}
\tag{20}
$$

$$
C_1 + C_2 = \mathbb{E}_{q(\mathbf{x}_{0:T})}\left[\log q\left(\mathbf{x}_0\right) - \sum_{t=1}^{T}\log q\left(\mathbf{x}_0\mid\mathbf{x}_t\right)\right].
\tag{21}
$$

Let $L = \log p_\theta(\mathbf{x}_0)$ be the model's log-likelihood for a data point $\mathbf{x}_0$, and let $K$ be its variational lower bound:

$$
K = \mathbb{E}_{q(\mathbf{x}_0)q(\mathbf{x}_{1:T}\mid\mathbf{x}_0)}\left[\log\frac{p_\theta\left(\mathbf{x}_{0:T-1}\mid\mathbf{x}_T\right)}{q\left(\mathbf{x}_{1:T}\mid\mathbf{x}_0\right)} + \log p\left(\mathbf{x}_T\right)\right].
\tag{22}
$$

Then, the following inequality chain holds:

$$
L \geq K = -\mathcal{L}_1 \geq -\mathcal{L}_2,
\tag{23}
$$

where $\mathcal{L}_1$ and $\mathcal{L}_2$ are defined as above.

The proof is based on two applications of Jensen's inequality to the log-likelihood.

**1. Proving $L \geq K$**

This is the standard variational lower bound for diffusion models, derived by applying Jensen's inequality:

$$
\begin{aligned}
\log p_\theta(\mathbf{x}_0) &= \log\int p_\theta(\mathbf{x}_{0:T})d\mathbf{x}_{1:T} \\
&= \mathbb{E}_{q(\mathbf{x}_0)}\left[\log\mathbb{E}_{q(\mathbf{x}_{1:T}\mid\mathbf{x}_0)}\left[\frac{p_\theta(\mathbf{x}_{0:T-1}\mid\mathbf{x}_T)p_\theta(\mathbf{x}_T)}{q(\mathbf{x}_{1:T}\mid\mathbf{x}_0)}\right]\right] \\
&\overset{(a)}{\geq} \mathbb{E}_{q(\mathbf{x}_{0:T})}\left[\log\frac{p_\theta(\mathbf{x}_{0:T-1}\mid\mathbf{x}_T)}{q(\mathbf{x}_{1:T}\mid\mathbf{x}_0)} + \log p_\theta(\mathbf{x}_T)\right] = K
\end{aligned}
\tag{24}
$$

Here, we assume $p_\theta(\mathbf{x}_T) \approx q(\mathbf{x}_T)$. This proves the first part of the inequality.

**2. Proving $K = -\mathcal{L}_1$**

$$
\begin{aligned}
K &= \mathbb{E}_{q(\mathbf{x}_{0:T})}\left[\log\prod_{t=1}^{T}\frac{p_\theta(\mathbf{x}_{t-1}\mid\mathbf{x}_t)}{q(\mathbf{x}_t\mid\mathbf{x}_{t-1})} + \log p_\theta(\mathbf{x}_T)\right] \\
&= \sum_{t=1}^{T}\mathbb{E}_{q(\mathbf{x}_{0:T})}\left[\log\frac{p_\theta(\mathbf{x}_{t-1}\mid\mathbf{x}_t)}{q(\mathbf{x}_t\mid\mathbf{x}_{t-1})}\right] + \mathbb{E}_{q(\mathbf{x}_{0:T})}\left[\log p_\theta(\mathbf{x}_T)\right]
\end{aligned}
\tag{25}
$$

From the derivations in (Sohl-Dickstein et al., 2015), the variational lower bound $K$ can be strictly rewritten in terms of KL divergences, which directly correspond to our $\mathcal{L}_1$. Specifically,

$$
\begin{aligned}
K = & -\sum_{t=2}^{T} \int d\mathbf{x}_0 d\mathbf{x}_T q\left(\mathbf{x}_0, \mathbf{x}_T\right) \cdot \mathrm{KL}\left(q\left(\mathbf{x}_{T-1} \mid \mathbf{x}_T, \mathbf{x}_0\right) \| p\left(\mathbf{x}_{T-1} \mid \mathbf{x}_T\right)\right) \\
& + H_q\left(\mathbf{x}_T \mid \mathbf{x}_0\right) - H_q\left(\mathbf{x}_1 \mid \mathbf{x}_0\right) - H_p\left(\mathbf{x}_T\right)
\end{aligned}
\tag{26}
$$

Since

$$
\begin{aligned}
H_q\left(\mathbf{x}_T \mid \mathbf{x}_0\right) - H_p\left(\mathbf{x}_T\right) = & \int_{\mathbf{x}_T} \int_{\mathbf{x}_0} q\left(\mathbf{x}_T \mid \mathbf{x}_0\right) q\left(\mathbf{x}_0\right) \log q\left(\mathbf{x}_T \mid \mathbf{x}_0\right) d\mathbf{x}_0 d\mathbf{x}_T \\
& - \int_{\mathbf{x}_T} \int_{\mathbf{x}_0} q\left(\mathbf{x}_t \mid \mathbf{x}_0\right) q\left(\mathbf{x}_0\right) d\mathbf{x}_0 \log p\left(\mathbf{x}_T\right) d\mathbf{x}_T \\
= & \int_{\mathbf{x}_T} \int_{\mathbf{x}_0} q\left(\mathbf{x}_T \mid \mathbf{x}_0\right) q\left(\mathbf{x}_0\right) \log \frac{q\left(\mathbf{x}_T \mid \mathbf{x}_0\right)}{p\left(\mathbf{x}_T\right)} d\mathbf{x}_0 d\mathbf{x}_T \\
= & \mathbb{E}_{p_{\text{data}}(\mathbf{x}_0)}\left[\mathrm{KL}\left(q\left(\mathbf{x}_T \mid \mathbf{x}_0\right) \| p\left(\mathbf{x}_T\right)\right)\right],
\end{aligned}
\tag{27}
$$

and $H_q\left(\mathbf{x}_1 \mid \mathbf{x}_0\right) = \mathbb{E}_{p_{\text{data}}} \mathbb{E}_{q(x_1 \mid \mathbf{x}_0)}\left[\log p_{0 \mid 1}\left(\mathbf{x}_0 \mid x_1\right)\right]$, then the above formula is equivalent to

$$
-\mathbb{E}_{\mathbf{x}_t \sim q_{T \mid 0}(\cdot \mid \mathbf{x}_0)}\left[D_{\mathrm{KL}}\left(\mathbb{P}_{\mathbf{x}_0}(\cdot \mid \mathbf{x}_t) \| \mathbb{P}^{\theta}(\cdot \mid \mathbf{x}_t)\right)\right] - D_{\mathrm{KL}}\left(q_{T \mid 0}(\cdot \mid \mathbf{x}_0) \| \pi\right),
\tag{28}
$$

which is equal to $\mathcal{L}_1$, according to (Lou et al., 2024).

**3. Proving $K \geq -\mathcal{L}_2$**

The derivation of $\mathcal{L}_2$ involves a second application of Jensen's inequality on $K$:

$$
\begin{aligned}
K = & \mathbb{E}_{q(\mathbf{x}_{0:T})}\left[\sum_{t=1}^{T} \log \frac{p_\theta\left(\mathbf{x}_{t-1} \mid \mathbf{x}_t\right)}{q\left(\mathbf{x}_t \mid \mathbf{x}_{t-1}\right)} + \log p\left(\mathbf{x}_T\right)\right] \\
= & \mathbb{E}_{q(\mathbf{x}_{0:T})}\left[\sum_{t=1}^{T} \log \left(\sum_{\tilde{\mathbf{x}}_0} q(\mathbf{x}_{t-1} \mid \mathbf{x}_t, \tilde{\mathbf{x}}_0) \tilde{p}_\theta(\tilde{\mathbf{x}}_0 \mid \mathbf{x}_t)\right)\right] + C_1 \\
\overset{(b)}{\geq} & \mathbb{E}_{q(\mathbf{x}_{0:T})}\left[\sum_{t=1}^{T} \sum_{\tilde{\mathbf{x}}_0} q(\tilde{\mathbf{x}}_0 \mid \mathbf{x}_{t-1}) \log \left(\frac{q(\mathbf{x}_{t-1} \mid \mathbf{x}_t)}{q(\tilde{\mathbf{x}}_0 \mid \mathbf{x}_t)} \tilde{p}_\theta(\tilde{\mathbf{x}}_0 \mid \mathbf{x}_t)\right)\right] + C_1 \\
= & \sum_{t=1}^{T} \mathbb{E}_{q(\mathbf{x}_t, \mathbf{x}_0)}\left[\log \tilde{p}_\theta(\mathbf{x}_0 \mid \mathbf{x}_t)\right] + C_1 + C_2 \\
= & -\mathcal{L}_2
\end{aligned}
\tag{29}
$$

In summary, $-\log p_\theta(\mathbf{x}_0) \leq \mathcal{L}_1 \leq \mathcal{L}_2$ holds.

## F  IMPLEMENTATION OF THE RESAMPLER

We define a resampler $S: \mathbb{R}^h \rightarrow \mathbb{R}^{L \times D}$, where $h$ is the length of the input vector, $L$ is the length of the sequence and $D$ is the hidden dimension of UniF$^2$ace. Specifically, we define a learnable hidden latent matrix:

$$
\mathbf{M}_0 \in \mathbb{R}^{L \times d}, \quad \mathbf{M}_0 = \text{LearnableParameter}
\tag{30}
$$

where $d$ is the hidden dimension of the resampler. Its process involves:

1. Project the noise embedding $\mathbf{x} \in \mathbb{R}^h$ via

$$
\mathbf{H} = \mathbf{x} \mathbf{W}_{\text{in}} \in \mathbb{R}^{1 \times d}
\tag{31}
$$

2. Iteratively refine the latent matrix through $T$ layers, sucn as the $l$-th layer:

$$\mathbf{M}'_l = \mathbf{M}_{l-1} + \text{MHA}\left(\mathbf{M}_{l-1}, \text{Concat}(\mathbf{H}, \mathbf{M}_{l-1})\right) \tag{32}$$

$$\mathbf{M}_l = \mathbf{M}'_l + \text{FFN}(\mathbf{M}'_l) \tag{33}$$

where MHA denotes the Multi-Head Attention mechanism, FFN denotoes the Feed-Forward Network. In MHA, the query, key, and value are denoted as:

$$Q_l = M_{l-1} W_Q^{(l)} \tag{34}$$

$$K_l = [H; M_{l-1}] W_K^{(l)} \tag{35}$$

$$V_l = [H; M_{l-1}] W_V^{(l)} \tag{36}$$

3. Project the final latent to the output space:

$$\mathbf{Y} = \text{LayerNorm}(\mathbf{M}_T \mathbf{W}_{\text{out}}) \in \mathbb{R}^{L \times D} \tag{37}$$

This enables adaptive fusion of input vector into sequence features through learned latent queries.

## G    TRAINING COST DETAILS

For analysis of training cost, we define the "simplest baseline" as a model trained directly without any of our proposed methods. As shown in Table 11, the detailed cost breakdown is as follows: (1) **D3Diff**: The introduction of D3Diff, a novel loss function, results in a modest +5.8% increase in training overhead, while achieving substantial improvements in generation performance as measured by VQAscore and VLM-score. (2) **Multi-level MoEs**: The multi-level MoEs expands parameters from 1.3B to 1.8B, resulting in a 40.7% increase in training cost. However, this is a worthwhile trade-off as the overhead scales nearly linearly with the parameter growth and, more importantly, delivers substantial improvements in both generation and understanding capabilities.

Table 11: Ablations of training cost vs. performance.

| Method | Params | GPU Hours | Generation | | Understanding | |
|---|---|---|---|---|---|---|
| | | | VQAscore↑ | VLM-score↑ | Desc↑ | Conv↑ |
| Baseline | 1.3B | 117.4 | 0.855 | 79.698 | 4.958 | 6.037 |
| UniF$^2$ace (w/ D3Diff) | 1.3B | 124.3 | 0.878 | 84.432 | 4.988 | 6.031 |
| UniF$^2$ace (w/ MoEs) | 1.8B | 165.2 | 0.879 | 85.993 | 5.885 | 6.427 |
| UniF$^2$ace (Full) | **1.8B** | **175.3** | **0.894** | **88.049** | **6.023** | **6.532** |

## H    PERFORMANCE ON THE NOISY INPUTS

Evaluating model robustness under degraded conditions is paramount for transitioning Multimodal Large Language Models (MLLMs) from idealistic, controlled environments to unpredictable real-world applications. However, given the scarcity of datasets containing authentic, naturally occurring facial noise, we employed synthetic noise simulations to rigorously assess performance stability. Specifically, we injected Gaussian noise into the RGB space with a mean ($\mu$) of 0 and a standard deviation ($\sigma$) of 25. This controlled degradation effectively emulates prevalent real-world artifacts—such as motion blur, environmental shadows, and partial occlusions—while ensuring that fundamental facial identity and structural features remain discernible for analysis.

We conducted a comparative analysis of face understanding performance for the aforementioned noisy inputs. Following the evaluation metrics established in the paper, we assessed both Description (Desc) capabilities-i.e., Captioning and Conversation (Conv) capabilities—i.e., Visual Question Answering (VQA). The experimental results, as shown in Table 12, unequivocally demonstrate that UniF$^2$ace exhibits strong robustness to input noise, achieving the best overall understanding performance across the majority of evaluation metrics on degraded face images.

## I    THE USE OF LARGE LANGUAGE MODELS (LLMS)

LLMs were used solely to aid in writing and polishing the text (e.g., improving clarity and grammar), with all outputs verified by the authors.

Table 12: Performance comparison under synthetic noisy inputs.

| Model | Params | Desc-GPT ↑ | Conv-GPT ↑ | Desc-DS ↑ | Conv-DS ↑ |
|---|---|---|---|---|---|
| Qwen2-VL (Wang et al., 2024a) | 7B | 5.29 | 4.76 | 6.03 | 5.84 |
| Qwen2.5-VL (Bai et al., 2025) | 3B | 5.38 | **5.61** | 6.21 | 6.14 |
| UniF$^2$ace (Ours) | **1.8B** | **5.43** | 5.58 | **6.37** | **6.29** |

| SDXL | TokenFlow | OmniFlow | Show-o | UniF$^2$ace(Ours) |
|---|---|---|---|---|

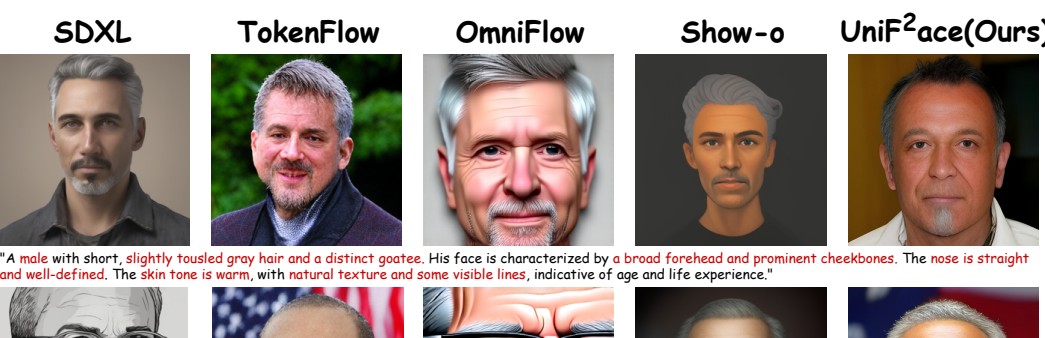

"A male with short, slightly tousled gray hair and a distinct goatee. His face is characterized by a broad forehead and prominent cheekbones. The nose is straight and well-defined. The skin tone is warm, with natural texture and some visible lines, indicative of age and life experience."

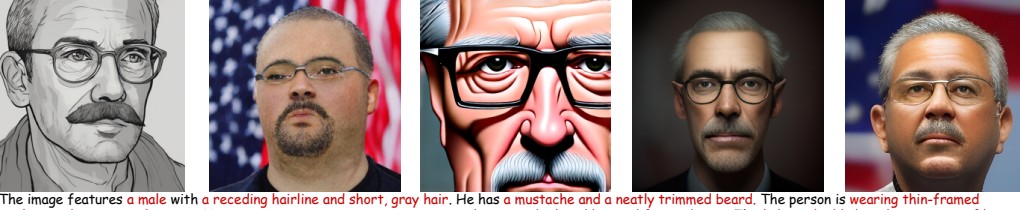

"The image features a male with a receding hairline and short, gray hair. He has a mustache and a neatly trimmed beard. The person is wearing thin-framed eyeglasses that rest on his nose. His expression appears serious or contemplative, with closed lips and focused eyes. The lighting highlights the contours of his face, emphasizing the forehead and cheekbones. The background is slightly blurred, featuring an American flag, suggesting a formal or official setting. The overall composition conveys a sense of gravity and focus."

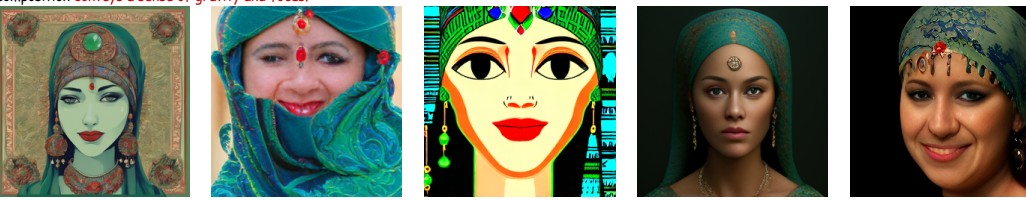

"The person in the image has a warm and friendly expression, characterized by a gentle smile that conveys happiness. They appear to be female and have light skin, with long hair partially covered by a patterned headscarf in shades of green and blue, adding elegance to their appearance. The decorative headpiece features a central red gem and dangling elements across the forehead. Their eyes are accentuated with dark eyeliner, and they have arched eyebrows that complement their facial features. The person is wearing lipstick, which enhances their smile, and they have a pointy nose. Additionally, they are wearing earrings that add a touch of sophistication. The lighting highlights the smooth texture of their skin, enhancing the natural beauty of their face."

Figure 8: More comparison of generated face images with other models. Fine-grained attributes are highlighted in the prompt.

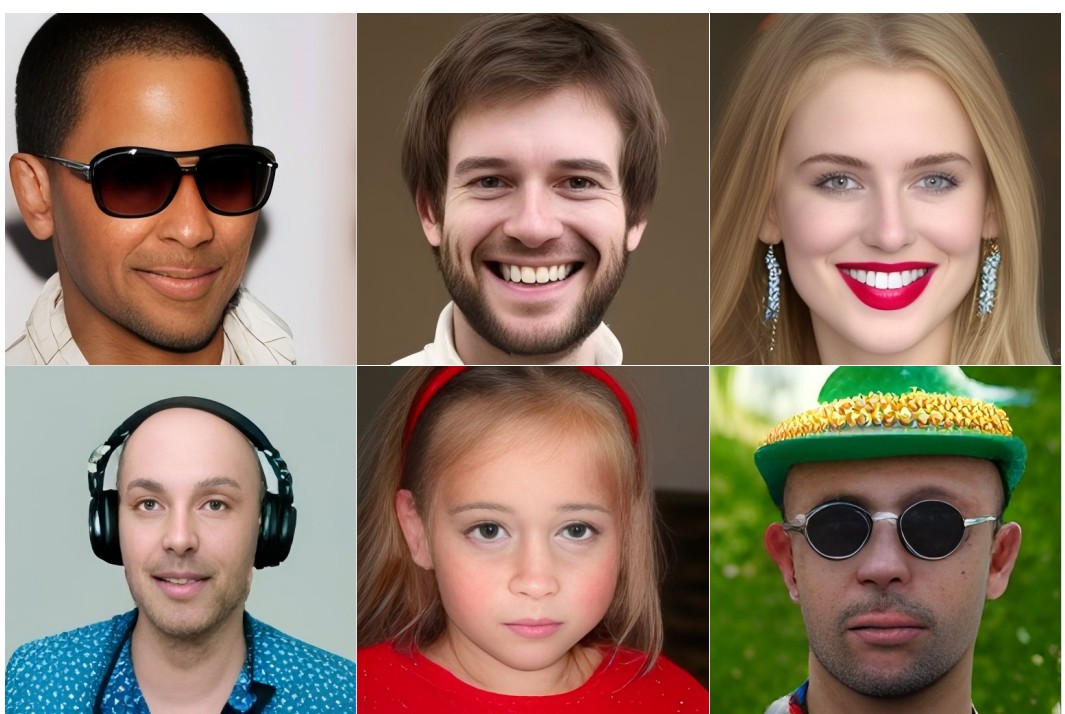

Figure 9: More face images generated by UniF²ace

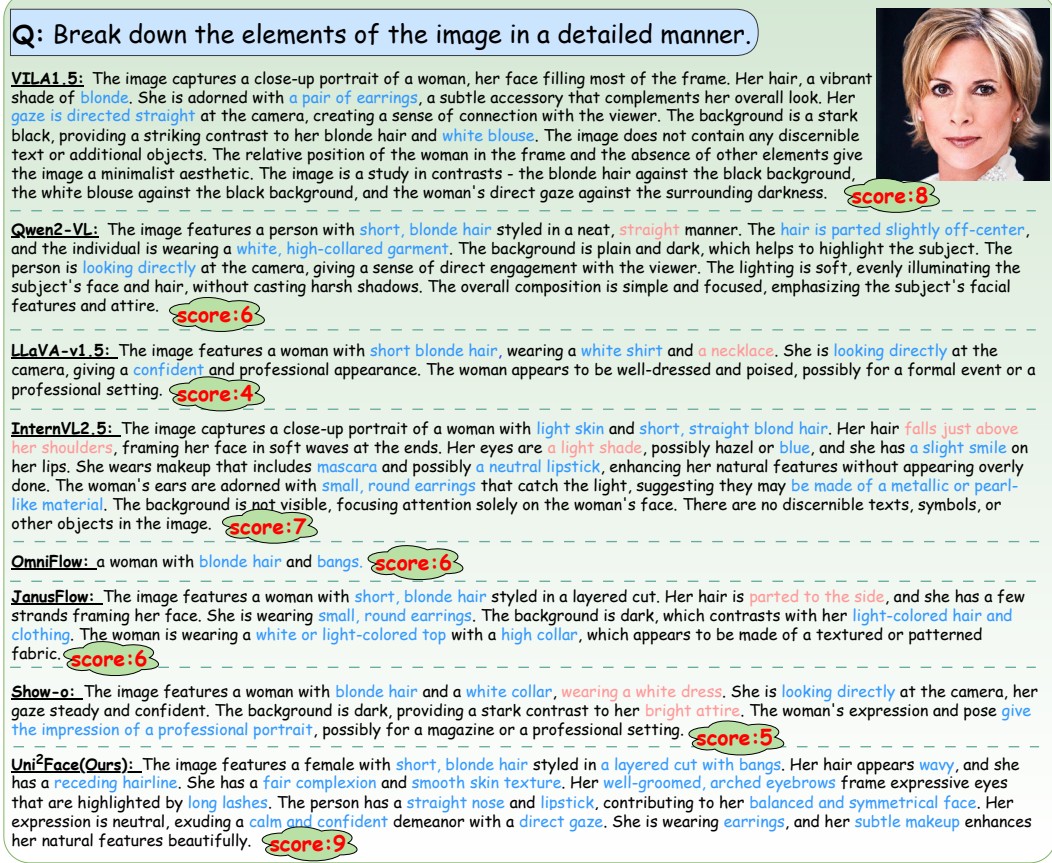

Figure 10: Comparison of captioning results and DeepSeek-v3-based scores. We highlight fine-grained attributes with blue and errors in answers with red.

> **Step1: Prompt for generating initial captions:**
>
> **User:** "In the image there is a person, describe the image in a paragraph giving detailed fine-grained attributes of the person face. [Note that the output is mixed with the captions given below: xxxxx]"
>
> **Step2: Prompt for generating fine-grained captios:**
>
> **User:** "Please combine the face caption you just replied to and the following features into one paragraph:
> Appearance: xxx, xxx, xxx, xxx......
> Action: xxx, xxx, xxx......
> Emotion: xxx
>
> **Step3: Prompt for generating fine-grained VQAs**
>
> **User:** You are an AI visual assistant, and you are seeing a face image. What you see are provided with a paragraph , describing the same image you are looking at. Answer all questions as you are seeing the image.
> Design a conversation between you and a person asking about this photo. The answers should be in a tone that a visual AI assistant is seeing the image and answering the question.
> Ask diverse questions and give corresponding answers.
> Questions cover as many face attributes as possible, such as hair, nose, eyes, mouth, ears, skin, eyebrows, adornment, and so on. Only include questions that have definite answers:
> (1) one can see the content in the image that the question asks about and can answer confidently;
> (2) one can determine confidently from the image that it is not in the image.
> Do not ask any question that cannot be answered confidently.
> Also include closed-ended questions that are relevant to the content in the image, for example, asking whether the person in the image has earrings, asking whether is the hair of the person in the image long or short, etc. Again, do not ask about uncertain details.
> Also include complex questions that are relevant to the content in the image, for example, asking about the action and emotion of the person in the image, asking to discuss about events happening in the image, etc. Again, do not ask about uncertain details.
> Provide detailed answers when answering complex questions. For example, give detailed examples or reasoning steps to make the content more convincing and well-organized. You can include multiple paragraphs if necessary.
> Please return the results in the following json format:
> Example:
> {"from": "human", "value": "Can you describe his eyes and eyebrows?"},
> {"from": "gpt", "value": "Certainly! His eyes are deep-set and expressive, and his bushy dark eyebrows complement them well, enhancing his expressive appearance."},
> {"from": "human", "value": "What color are her earrings?"},
> {"from": "gpt","value": "She wears gold earrings."},

Figure 11: Prompts for building dataset. The first and second prompts are to GPT-4o, while the last prompt is to GPT-4. In the first prompt, the content in "[]" is used only when the image data includes built-in captions, such as in the MM-CelebA-HQ dataset.

