# OpenReview forum: "UniF$^2$ace: A $\underline{Uni}$fied $\underline{F}$ine-grained $\underline{Face}$ Understanding and Generation Model"
_ICLR.cc/2026/Conference — ICLR 2026 Poster_

### Official Review · Reviewer_WDVe · 2025-10-30

**Soundness:** 3
**Presentation:** 4
**Contribution:** 3
**Rating:** 6
**Confidence:** 3

**Summary:**

The paper introduces UniF2ace, a model that unifies face understanding (captioning, VQA) and generation (text-to-image) in one framework. It combines a new Dual Discrete Diffusion (D3Diff) loss with a multi-level Mixture-of-Experts design, and introduces a large dataset (UniF2aceD-1M) with detailed captions and 1M VQAs. Experiments show it outperforms state-of-the-art models on both tasks, even against larger systems.

**Strengths:**

1. First unified model for fine-grained face understanding + generation
2. Novel D3Diff loss improves detail and fidelity
3. New large-scale dataset with rich annotations
4. Strong results across benchmarks, with thorough ablations

**Weaknesses:**

1. Limited discussion of ethical risks (deepfakes, misuse)
2. Dataset bias and annotation quality not fully addressed
3. Training appears complex and resource-heavy

**Questions:**

1. How balanced and unbiased is the dataset?
2. How well does the model handle real-world noisy inputs or domain shifts?
3. What is the computational/training cost compared to simpler baselines?

---

> ### Author Response · Authors · 2025-11-20
> **Author Response[1/2]**
>
> We sincerely thank you for taking the time to review our work and providing thoughtful feedback and constructive suggestions! Below, we address your comments in detail.
> ### **(1). The annotation quality and unbias of the dataset**
> Our dataset, UniF$^2$aceD-1M, was constructed via a rigorous three-step process designed to ensure high quality and mitigate bias (detailed in Appendix B). The resulting dataset features a rich average of 17.7 attributes per caption and demonstrates superior diversity compared to existing face-text datasets (Figure 4).
> + **Step-1, image collection**: We sourced high-resolution facial images from three independent datasets—CelebV-HQ[1], FFHQ[2], and MM-CelebA-HQ[3]. This multi-source strategy ensures high diversity and allows to inherit the statistically unbiased distributions that were validated in the original paper of these foundational datasets.
> + **Step-2, fine-grained caption building**: The creation of fine-grained captions relies on integrating general MLLM-generated(GPT-4o) descriptions with specific, high-fidelity attribute predictions. To ensure prediction veracity, we trained and employed classifiers that were rigorously pre-validated to meet stringent accuracy thresholds: over 93% for appearance, 87% for action, and 80% for emotion, with performance surpassing existing MLLMs.
> + **Step-3, VQAs creation**: We used GPT-4 to convert detailed captions into the final Vision Question-Answering (VQA) pairs.
> ### **(2). Handling real-world noisy inputs or domain shifts**
> 1. **Updated experiments for noisy inputs**: We thank the reviewer for this insightful question regarding real-world noisy inputs. Accurately modeling such noise is challenging, but we can evaluate robustness by applying standard noise types, such as Gaussian noise, to our test set. We believe this is a valuable analysis. We are currently conducting these experiments and will update our rebuttal and the final paper with the findings within the next 2-3 days.
> 2. **Performance when domain shifts**: We would like to direct the reviewer's attention to the **out-of-domain (OOD) evaluation** already presented in our initial submission. In these experiments, the test datasets were sourced from domains distinct from that of UniF$^2$aceD-1M. The results, detailed across two tables, demonstrate our model's strong generalization capabilities:
>    + **Generation Performance (Table 2)**: The results show that our model outperforms both generation-only models and Unified Multimodal Models (UMMs) of a similar parameter scale. Furthermore, its performance is comparable to that of significantly larger models, such as Flux.1-dev.
>    + **Understanding Performance (Table 4)**: The results indicate that our model also achieves highly competitive performance against both understanding-only models and other UMMs.
>
>
> [1] Zhu H, Wu W, Zhu W, et al. CelebV-HQ: A large-scale video facial attributes dataset[C]//European conference on computer vision. Cham: Springer Nature Switzerland, 2022: 650-667.
>
> [2] Karras T, Laine S, Aila T. A style-based generator architecture for generative adversarial networks[C]//Proceedings of the IEEE/CVF conference on computer vision and pattern recognition. 2019: 4401-4410.
>
> [3] Xia W, Yang Y, Xue J H, et al. Tedigan: Text-guided diverse face image generation and manipulation[C]//Proceedings of the IEEE/CVF conference on computer vision and pattern recognition. 2021: 2256-2265.

---

> ### Author Response · Authors · 2025-11-20
> **Author Response[2/2]**
>
> ### **(3). Modest Training Cost, Significant Performance Gains**
> For analysis of training cost, we define the "simplest baseline" as a model trained directly without any of our proposed methods. The detailed cost breakdown is as follows:
> + **D3Diff**: The introduction of D3Diff, a noval loss function results in a modest +5.8% increase in training overhead. But it achieves substantial improvements in generation performance, as measured by VQAscore and VLM-score.
> + **Multi-level MoEs**: The multi-level MoEs expands parameters from 1.3B to 1.8B, resulting in a 40.7% increase in training cost. However, this is a worthwhile trade-off. The overhead scales nearly linearly with the parameter growth and, more importantly, delivers substantial improvements in both generation and understanding capabilities.
> |Method|Parameters Size| GPU Hours|VQAscore↑|VLM-score↑|Desc↑|Conv↑|
> |:---|:---:|:---:|:---:|:---:|:---:|:---:|
> |Baseline|1.3B|117.4|0.855|79.698|4.958|6.037|
> |UniF$^2$ace(only with D3Diff)|1.3B(+0.0%)|124.3(+5.8%)|0.878|84.432|4.988|6.031|
> |UniF$^2$ace(only with Multi-level MoEs) |1.8B(+38.5%)|165.2(+40.7%)|0.879|85.993|5.885|6.427|
> |UniF$^2$ace(with D3Diff and Multi-level MoEs)|1.8B(+38.5%)|175.3(+49.3%)|**0.894**|**88.049**|**6.023**|**6.532**|
>
> ### **(4). Discussion of ethical risks**
> We thank the reviewer for raising this critical concern regarding ethical risks for misuse. We implemented two measures to underscore our commitment to ethical AI development.
> 1. **Check for hallucinations**: We enriched existing public datasets with a new layer of secondary annotations. To ensure the integrity of this layer, we implemented a rigorous protocol. The annotation workflow was designed to actively mitigate systemic bias and all LLM outputs underwent meticulous manual verification to eliminate hallucinations and factual errors.
> 2. **Commitment to responsible AI**: To ensure responsible use, we are releasing our model and dataset with safeguards. They are restricted to non-commercial research under a license consistent with source datasets, and are accompanied by terms of use that explicitly forbid malicious applications, such as deepfake generation.

---

> ### Author Response · Authors · 2025-11-24
> **Author Response: UniF$^2$ace also works well on the noisy inputs**
>
> Due to the difficulty in obtaining realistic, naturally noisy human face datasets, we conducted experiments involving the simulation of added noise (synthetic noise). In our experimental setup, we simulated noisy inputs prevalent in real-world scenarios by applying Gaussian noise to the face images. Specifically, we introduced noise in the RGB space with a mean of 0 and a standard deviation of 25. It successfully exhibited real-world degradation factors such as blurring, occlusion, and shadows, while the identifiability of the human face remained largely preserved.
>
> We conducted a comparative analysis of face understanding performance for the aforementioned noisy inputs. Following the evaluation metrics established in the paper, we assessed both Description (Desc) capabilities-i.e., Captioning and Conversation (Conv) capabilities—i.e., Visual Question Answering (VQA). The experimental results unequivocally demonstrate that UniF$^2$ace exhibits strong robustness to input noise, achieving the best overall understanding performance across the majority of evaluation metrics on degraded face images.
>
> | Model | Type | Params | Desc-GPT $\uparrow$ | Conv-GPT $\uparrow$ | Desc-DS $\uparrow$ | Conv-DS $\uparrow$ |
> | :--- | :--- | :--- | :--- | :--- | :--- | :--- |
> | Qwen2-VL | Und. Only | 7B | 5.29 | 4.76 | 6.03 | 5.84 |
> | Qwen2.5-VL | Und. Only | 3B | 5.38 | **5.61** | 6.21 | 6.14 |
> | UniF$^2$ace | Gen. & Und. | **1.8B** | **5.43** | 5.58 | **6.37** | **6.29** |

---

### Official Review · Reviewer_PpVm · 2025-11-02

**Soundness:** 3
**Presentation:** 3
**Contribution:** 3
**Rating:** 8
**Confidence:** 3

**Summary:**

This paper presents UniF2ace, a unified multimodal model (UMM) that integrates fine-grained face analysis and synthesis within a single framework. Additionally, the paper introduces UniF2aceD-1M, a new large-scale dataset featuring 130K high-resolution images and 1M fine-grained Visual Question Answering (VQA) pairs, annotated with an unprecedented average of 17.7 details across 46 distinct attributes.

**Strengths:**

The paper makes a strong theoretical contribution through the proposed Dual Discrete Diffusion (D3Diff) loss, which is formally derived to offer a tighter upper bound to the Negative Log-Likelihood (NLL) compared to conventional masked generative losses, moving the work beyond purely empirical engineering.

Architecturally, the model is innovative, employing a multi-level grouped Mixture-of-Experts (MoE) design with distinct token-level and sequence-level routing. This structure, which utilizes specialized CLIP and Face experts, provides both interpretable specialization and computational efficiency via sparsity.

A substantial dataset contribution is also provided with UniF2aceD-1M. This resource is highly valuable due to its diversity, high quality, and rich dual supervision (captioning and VQA) across fine-grained face attributes, significantly improving upon predecessors like FFHQ-Text.

**Weaknesses:**

While the ablation studies (e.g., D3Diff in Tab. 5, MoE in Tabs. 6–8) are technically sound, the paper provides limited practical intuition for the observed improvements. Specifically, the rationale behind optimal hyperparameter choices, such as why a Top-K=2 configuration consistently outperforms others, is presented only as an empirical result. A deeper, theoretical exploration of the hyperparameter sensitivity and architectural dynamics would greatly enhance the contribution.

Given the model's focus on generating and analyzing human facial data, the ethical discussion is critically minimal. The paper must include a dedicated and robust discussion addressing potential issues related to bias, privacy, and misuse, such as the generation of deepfakes or facial profiling. Furthermore, the dataset sourcing and consent must be clarified. Since UniF2aceD-1M is sourced from web-scraped and public datasets (FFHQ, CelebV-HQ, MM-CelebA-HQ), the authors must provide explicit details on data licensing, subject consent, and any filtering mechanisms used.

The current empirical evaluation lacks diversity. All benchmarks rely on ideal, celebrity-style, or synthetic datasets. The paper fails to test the model’s robustness and generalizability across real-world conditions, such as faces with occlusion, low-light conditions, or poor image quality. This limits confidence in the model's real-world applicability.

**Questions:**

The ablation studies clearly show that a Top-K=2 configuration is optimal for the MoE routing. Is there a deeper, non-empirical reason for this? Can the authors provide a more intuitive or theoretical explanation for why using two experts performs better than using a single expert or dispersing the load among more (e.g., K=4), especially in the context of balancing semantic and identity embeddings?

The dataset UniF2aceD-1M combines CelebV-HQ, FFHQ, and MM-CelebA-HQ, among others. Could you elaborate on the data licensing and consent aspects, especially for identifiable human faces? Were any filtering, de-identification, or bias auditing steps performed (e.g., balancing across demographics)? Will dataset release include ethical usage guidelines or restricted licensing to prevent misuse (e.g., deepfakes, impersonation)?

The dataset is described as “fine-grained” but primarily based on high-quality celebrity-style images. How diverse is UniF2aceD-1M in terms of age, ethnicity, lighting, and pose variation? Do you observe any performance degradation when tested on non-celebrity, low-quality, or occluded faces?

---

> ### Author Response · Authors · 2025-11-20
> **Author Response[1/2]**
>
> Thank you for taking the time to provide a very detailed and constructive review of our submission! Below, we address your questions and comments individually, incorporating clarifications and additional insights where relevant.
> ### **(1). Top-2 vs. Top-4: A performance analysis in MoEs**
> 1. **Superiority of the Top-4**: We would like to respectfully clarify that the Top-2 strategy does not comprehensively outperform the Top-4 strategy, as shown in Table 7. In fact, the Top-4 strategy demonstrates better performance on a majority of the metrics, including VQAscore and FID for generation and the Conversation (Conv) metric for understanding. Furthermore, their performance on the VLMScore metric is highly comparable (88.049 vs. 87.795).
> 2. **Advantage analysis of Top-2 in the description task**: The one metric where Top-2 shows an advantage is in the Description (Desc) task, and our analysis points to a specific reason. The Desc task evaluates general face captioning, which involves describing features common to different faces. This encourages the MoE to converge on a specialized routing path, with Figure 6 (bottom left) confirming the dominant activation of the semantic CLIP Expert. For such a specialized task, the greater diversity of Top-4, combined with the MoE's load-balancing loss, can inadvertently increase optimization difficulty. This results in slightly lower performance compared to the more constrained Top-2 strategy. In contrast, the Conversation (Conv) task, which involves diverse questions about various facial regions, requires a wider range of experts. Consequently, it benefits from the more flexible routing of Top-4, leading to better performance.
> 3. **Top-2 is a better trade-off**: While the Top-4 strategy outperforms Top-2 on some metrics, the overall performance of Top-2 remains highly comparable. Crucially, Top-4 doubles the inference cost in the Transformer's FFN layer, making Top-2 a much better trade-off between performance and efficiency.
> ### **(2). Data licensing and consent aspects**
> We thank the reviewer for concern and reminding us of the important aspects of data licensing and consent.
> 1. **Review of source licenses**: To this end, we have conducted a thorough review of the licenses for all source datasets (CelebV-HQ[1], FFHQ[2], and MM-CelebA-HQ[3]) and verified their allowance for non-commercial research and educational applications.
> 2. **Manual check for LLM**: It is also important to note that our work constitutes a secondary annotation based on these existing, publicly available datasets. During our annotation process, we took great care to avoid introducing any form of bias or preference. Furthermore, all contributions from the LLM were manually checked by our team to prevent the introduction of hallucinations.
> 3. **Preventing misuse with explicit state**: Upon the release of our dataset, we will explicitly state these details and ensure that it inherits the licenses of source datasets to prevent potential misuse.
>
>
> [1] Hao Zhu, et al. 2022, CelebV-HQ: A large-scale video facial attributes dataset.
>
> [2] Tero Karras, et al. 2019, A style-based generator architecture for generative adversarial networks.
>
> [3] Weihao Xia, et al. 2021, Tedigan: Text-guided diverse face image generation and manipulation.

---

> ### Author Response · Authors · 2025-12-03
> **Author Response[2/2]**
>
> ### **(3). Diversity of dataset and new experments on non-celebrity faces**
> We sincerely thank the reviewer for their constructive suggestions for improving our dataset and evaluation.
>
> 1. **Motivation for high-quality images used**: High-quality facial datasets, where detailed attributes are clearly visible, have proven to be highly efficient and effective for training both understanding and generative models, as supported by [1-3]. Motivated by this, we dedicated our efforts to collecting and curating such a dataset to facilitate the training of unified multimodal models and to provide a valuable open-source contribution to the community.
>
> 2. **Diverse distribution of UniF$^2$aceD-1M**: Utilizing a three-step pipeline (detailed in Appendix B), we constructed UniF$^2$aceD-1M, a dataset where each image is annotated with an average of 17.7 fine-grained attributes. These attributes encompass a wide range of categories, including gender, facial features, accessories, hairstyle, face shape, and more. Specifically, UniF$^2$aceD-1M exhibits superior diversity and balance, achieved through a strategic mixture of multi-source datasets. In terms of ethnicity, the dataset comprises approximately 34% Caucasian, 26% Asian, 24% African, and 16% Latino populations. The age demographics span 18% minors (<18), 61% adults (18--55), and 21% seniors (>55). Furthermore, regarding environmental attributes, pose variations are distributed across yaw angles with approximately 45% frontal, 36% semi-profile, and 19% profile views. Finally, concerning lighting conditions, challenging in-the-wild scenarios (e.g., dim light, backlight) account for 61% of the images, while indoor scenes constitute the remaining 39%.
>
> 3. **UniF$^2$ace also works well on the non-celebrity inputs**: We benchmarked our model against Qwen2-VL(7B) and Qwen2.5-VL(3B) using 3,000 random samples from FLIP-80M[4] to assess robustness in face understanding. Distinguished from standard celebrity-centric datasets, FLIP-80M (containing 80M+ pairs) captures the complexity of real-world data, including web-scraped and AI-generated content. The dataset presents substantial challenges regarding inconsistent lighting, blur, and unconstrained compositions, consisting largely of low-resolution, non-celebrity images.
>
>    | Model | Type | Params | Desc-GPT $\uparrow$ | Desc-DS $\uparrow$ |
>    | :--- | :--- | :--- | :--- | :--- |
>    | Qwen2-VL | Und. Only | 7B | 4.97 | 5.13 |
>    | Qwen2.5-VL | Und. Only | 3B | 5.30 | 5.51 |
>    | UniF$^2$ace | Gen. & Und. | **1.8B** | **5.54** | **5.77** |
>
> [1] Xing B, Yu Z, Liu X, et al. Emo-llama: Enhancing facial emotion understanding with instruction tuning[J]. arXiv preprint arXiv:2408.11424, 2024.
>
> [2] Dai D, Jia M, Zhou Y, et al. Face-makeup: Multimodal facial prompts for text-to-image generation[J]. arXiv preprint arXiv:2501.02523, 2025.
>
> [3] Chen J, Xu Z, Pan X, et al. Blip3-o: A family of fully open unified multimodal models-architecture, training and dataset[J]. arXiv preprint arXiv:2505.09568, 2025.
>
> [4] Li Y, Hou X, Dezhi Z, et al. Flip-80m: 80 million visual-linguistic pairs for facial language-image pre-training[C]//Proceedings of the 32nd ACM International Conference on Multimedia. 2024: 58-67.

---

### Official Review · Reviewer_7WgR · 2025-11-03

**Soundness:** 4
**Presentation:** 4
**Contribution:** 4
**Rating:** 8
**Confidence:** 4

**Summary:**

This paper proposes a unified multimodal large language model framework for facial understanding and generation in an auto-regressive plus diffusion-based style. The paper develops a novel multi-level, grouped mixture-of-expert-based architecture, incorporating the advantages of semantic and identity features from CLIP and facial experts. To facilitate fine-grained facial understanding and generation, a fine-grained facial image-caption dataset, uniF^2aceD-1M, containing large-scale visual question-answering pairs across diverse facial attributes, is simultaneously constructed to train the model. Additionally, a novel dual discrete diffusion (D3Diff) loss is proposed to enhance the quality of generated images.

**Strengths:**

1. The paper is well-written and easy to follow.
2. The paper is the first one to propose a unified multimodal framework for both facial understanding and generation, supervised by a compound loss for next-token textual generation and diffusion-based image generation.
3. A novel architecture, based on a multi-level grouped mixture of experts, incorporating semantic and identity features, is proposed to enhance the model's understanding of fine-grained facial attributes.
4. A novel dual discrete diffusion (D3Diff) loss with a theoretical guarantee is proposed to improve the generation quality further.
5. Thorough experimental results demonstrate the effectiveness of the proposed method in both facial understanding and generation benchmarks.

**Weaknesses:**

1. The evaluation of fine-grained facial understanding should include the result in a challenging benchmark, FaceBench, developed for fine-grained facial understanding across over 210 facial attributes.
2. Concern about the fairness in evaluation. Were other methods pre-trained on the UniF^2ace training set before evaluation, or were they evaluated in a zero-shot manner? If they are evaluated in a zero-shot manner, their performances are certainly below the upper bound compared to the proposed method, which is pretrained on fine-grained facial understanding and generation data.
3. It is better and strongly recommended to include the user study for both facial understanding and generation.

FaceBench: A Multi-View Multi-Level Facial Attribute VQA Dataset for Benchmarking Face Perception MLLMs. CVPR 2025.

**Questions:**

1. Do the authors have any plan to scale up the model size (e.g., larger than 7B) to boost the performance further?
2. Is there any open-source plan for the training, inference, evaluation codes, and the checkpoints?

---

> ### Author Response · Authors · 2025-11-20
> **Author Response**
>
> Thank you so much for your detailed and constructive feedback! We sincerely appreciate the time and effort you have taken to thoroughly engage with our work and provide us with such thoughtful suggestions for improvement.
>
> ### **(1). About scaling up the model size**
> 1. **Our 1.8B model outperforming 7B+ giant**: Due to computational constraints, we have currently trained a relatively small version of our model (1.8B). Despite its modest scale, UniF$^2$ace demonstrates superior performance against significantly larger models, as highlighted in red across our comparative experiments (Tables 1-4). For instance, in understanding tasks, it outperforms 7B+ models like Qwen2-VL (7B)[1], InternVL2.5 (8B)[2] and TokenFlow(7B)[3]. In generation tasks, it outperforms Flux.1-dev (12B)[4] and TokenFlow(7B)[3]. These results validate the effectiveness of UniF$^2$ace in the domain of fine-grained face analysis.
>
> 2. **Future work: scaling to 7B+**: Acknowledging the strong scaling properties demonstrated by recent Unified Multimodal Models (UMMs)[6, 7, 8], our future work will focus on scaling UniF$^2$ace to 7B parameters and beyond as more computational resources become available, aiming to build a more powerful foundation model for the fine-grained face.
>
> ### **(2). Open-source plan**
> The complete code for training and inference is included in the initial supplementary materials. Upon acceptance of this paper, we will make all project components, including code and models, publicly available.
>
> ### **(3). Evaluation in FaceBench**
> 1. **Updating related work section**: We apologize for overlooking the recent FaceBench[5] benchmark, a relevant work in fine-grained face scenarios. We will be sure to add a citation and discussion of this work in our revised paper.
>
> 2. **Comprehensive evaluation**: Regarding the broader concerns about our evaluation, we would like to highlight two key aspects of our methodology.
>    + **Fine-grained facial evaluation**: As detailed in Section 4.1, we constructed a comprehensive test set where each caption contains an average of 17.7 facial attributes spanning appearance, action, and emotion. It includes categories such as hair, beard, accessories, skin conditions, lipstick and so on (as detailed in Figure 4). We believe this dataset offers greater representativeness and diversity compared to many existing face-related benchmarks.
>    + **OOD evalutaion**: We explicitly tested our model on out-of-domain (OOD) datasets to assess its generalization and robustness, with the results presented in Tables 2 and 4.
>
> We hope this explanation helps to alleviate your concerns regarding the thoroughness of our evaluation.
>
>
> ### **(4). About the fairness in evaluation**
> We thank the reviewer for pointing out this ambiguity, and we apologize that our description of the experimental setup was not sufficiently clear. To clarify, for a fair comparison, all baseline models included in our experiments were fine-tuned on our UniF$^2$aceD-1M training dataset using their respective official code. We will make this point clearer in the revised paper.
>
> ### **(5). Promising user study**
> We thank the reviewer for this valuable suggestion. We agree that a user study would provide a more robust evaluation of fine-grained facial results. This is particularly true for generation tasks, where direct human feedback is invaluable for assessing the perceptual quality and user preference of the generated faces. While we were unable to conduct such a study for this submission due to limitations in human resources, we plan to incorporate a comprehensive user evaluation of our visual comparison results in future work.
>
>
> [1] Wang P, Bai S, Tan S, et al. Qwen2-vl: Enhancing vision-language model's perception of the world at any resolution[J]. arXiv preprint arXiv:2409.12191, 2024.
>
> [2] Chen Z, Wang W, Cao Y, et al. Expanding performance boundaries of open-source multimodal models with model, data, and test-time scaling[J]. arXiv preprint arXiv:2412.05271, 2024.
>
> [3] Qu L, Zhang H, Liu Y, et al. Tokenflow: Unified image tokenizer for multimodal understanding and generation[C]//Proceedings of the Computer Vision and Pattern Recognition Conference. 2025: 2545-2555.
>
> [4] Black Forest Labs. Flux. https://github.com/black-forest-labs/flux, 2024.
>
> [5] Wang X, Ma X, Hou X, et al. FaceBench: A Multi-View Multi-Level Facial Attribute VQA Dataset for Benchmarking Face Perception MLLMs[C]//Proceedings of the Computer Vision and Pattern Recognition Conference. 2025: 9154-9164.
>
> [6] Team C. Chameleon: Mixed-modal early-fusion foundation models[J]. arXiv preprint arXiv:2405.09818, 2024.
>
> [7] Chen X, Wu Z, Liu X, et al. Janus-pro: Unified multimodal understanding and generation with data and model scaling[J]. arXiv preprint arXiv:2501.17811, 2025.
>
> [8] Chen J, Xu Z, Pan X, et al. Blip3-o: A family of fully open unified multimodal models-architecture, training and dataset[J]. arXiv preprint arXiv:2505.09568, 2025.

---

> > ### Comment · Reviewer_7WgR · 2025-11-21
> > **Official Comment**
> >
> > Thanks for the clear and convincing responses from the authors. The issues have been well addressed. The proposed method makes a significant contribution to the field of facial understanding and generation. This work provides an excellent infrastructure and framework to the community, inspiring researchers in related topics to develop more advanced approaches.

---

> > > ### Author Response · Authors · 2025-11-24
> > > **Appreciation for the Reviewer's Positive Recognition**
> > >
> > > We sincerely thank the reviewer for the insightful comments and positive recognition of our work. The suggestions provided are highly valuable and have significantly contributed to enhancing the quality and clarity of the paper.

---

### Official Review · Reviewer_9L5B · 2025-11-03

**Soundness:** 3
**Presentation:** 3
**Contribution:** 2
**Rating:** 4
**Confidence:** 4

**Summary:**

This paper presents a unified multimodal model that integrates fine-grained face understanding and generation into a single framework. It introduces a dual discrete diffusion loss and multi-level grouped MoE architecture. Which combines token-level and sequence-level expert routing to preserve semantic and identity features during multimodal representation learning. Additionally, the authors introduce a large-scale dataset (130K image caption pairs and 1M VQAs) designed for fine-grained facial attribute learning. Experiments demonstrate state-of-the-art performance on both generation and understanding tasks.

**Strengths:**

1.	The proposed D3Diff loss provides a principled unification of score-based and masked generative modeling.
2.	This paper proposes a multi-level grouped MoE structure effectively separates semantic and identity embeddings,.

**Weaknesses:**

1.	The authors should more clearly justify the necessity of a unified model for fine-grained face understanding and generation. The current motivation might not be accepted by the community, as expert models specifically designed for face understanding and generation already handle these tasks well.
2.	The study’s exploration of fine-grained face understanding appears limited, primarily focusing on attributes such as hair and beard. This narrow scope raises concerns about the generality and depth of the proposed approach. Furthermore, the paper overlooks several recent works that specifically address face understanding, such as:
[1] FaceXBench Evaluating Multimodal LLMs on Face Understanding
[2] FaceInsight: A Multimodal Large Language Model for Face Perception
[2] FaVChat: Unlocking Fine-Grained Facial Video Understanding with Multimodal Large Language Models
3.	The motivation for sequence-level MoE layers and token-level MoE layers is not very clear. Why is a token-level MoE layer needed, and what role does it play in face understanding or generation tasks?

**Questions:**

Please refer to the Weaknesses.

---

> ### Author Response · Authors · 2025-11-20
> **Author Response [1/2]**
>
> We sincerely thank you for taking the time to provide detailed and constructive feedback on our submission. Your comments are highly valuable and have significantly helped us identify areas for improvement in our work. Below, we address your suggestions and concerns.
>
> ### **(1). Justification of the necessity for face UMM**
> Our proposed fine-grained face Unified Multimodal Model (UMM) offers distinct advantages over task-specific expert models:
>
> 1. **Understanding-enhancing Generation**: The UMM uniquely utilizes its comprehension abilities (e.g., via thinking or recaptioning, as demonstrated in prior work[1, 2]) to rewrite users' prompts, leading to superior generation quality. This is a feature absent in generation-only expert models.
>
> 2. **Promising advanced multi-subject editing**: Leveraging in-context learning, the UMM's architecture achieves a superior fusion of cross-modal information, significantly outperforming specialized generative models like MMDiT[3]. Through superior cross-modal fusion, the face UMM enables more precise, identity-preserving edits. Furthermore, a key focus of our future work will be to leverage the UMM's scalable sequence-based design for complex multi-subject image composition. This research direction aims to address a critical limitation in existing systems and challenge the capabilities of state-of-the-art closed-source models (e.g., Seedream 4.0[4], Nano Banana[5]).
>
> 3. **Deployment Efficiency**: By unifying diverse functionalities into a single architecture, the Face UMM streamlines the deployment pipeline on resource-constrained edge devices (e.g., mobile phones, wearables). This approach eliminates the need to optimize multiple disparate models, significantly reducing both engineering overhead and computational costs. Let us illustrate these two benefits with a specific example for each.
>    + **Reduced engineering overhead**: Deploying models to edge devices typically requires significant effort from infrastructure engineers for optimizations like model compression and quantization. A unified architecture can halve this effort compared to maintaining and optimizing two distinct models.
>    + **Lower computational cost**: Consider a scenario with two independent expert models of 3B parameters each. Our unified model matches this capability within a single 3B parameter budget. During on-device inference, the unified model consumes a smaller memory footprint (VRAM). This efficiency stems from avoiding the redundant overhead associated with loading and managing two separate model instances, such as duplicated framework libraries and fragmented memory allocation.
>
> Collectively, these strengths underscore the importance of our Face UMM in the pursuit of robust and efficient human-centric AGI.
>
> [1] Cao S, Chen H, Chen P, et al. Hunyuanimage 3.0 technical report[J]. arXiv preprint arXiv:2509.23951, 2025.
>
> [2] Deng C, Zhu D, Li K, et al. Emerging properties in unified multimodal pretraining[J]. arXiv preprint arXiv:2505.14683, 2025.
>
> [3] Esser P, Kulal S, Blattmann A, et al. Scaling rectified flow transformers for high-resolution image synthesis[C]//Forty-first international conference on machine learning. 2024.
>
> [4] Seedream T, Chen Y, Gao Y, et al. Seedream 4.0: Toward next-generation multimodal image generation[J]. arXiv preprint arXiv:2509.20427, 2025.
>
> [5] Google. Nano banana, 2025. URL https://developers.googleblog.com/en/introdu
> cing-gemini-2-5-flash-image.

---

> ### Author Response · Authors · 2025-11-20
> **Author Response [2/2]**
>
> ### **(2). Diverse fine-grained face study**
> 1. **Exploration of fine-grained face**: By a meticulously designed, three-step pipeline(as detailed in Appendix B), we constructed UniF²aceD-1M, a dataset where each image has an average of 17.7 fine-grained facial attributes from a pool of over 150 types. As illustrated in Figure 4 (right), the hair and beard comprise only 10 attributes and our dataset includes categories such as accessories, skin, face shape, lipstick and so on. Further attributes, such as eye color, eye bags, talking statu and so on, are not fully listed due to space constraints, but an example is provided in Figure 7 (right).
> 2. **Diverse benchmark**: To clarify the comprehensiveness of evaluation: UniF2aceD-1M test set covers a wide spectrum of attributes (~17.7 per image) and we confirmed robustness on OOD well-established datasets like FFHQ-Text, MM-CelebA-HQ-Text and CelebV-Text (Table 4). While we did not evaluate directly on FaceXBench in our initial submission, we did ensure a rigorous comparison by benchmarking our model against key competitors in FaceXBench, such as Qwen2.5-VL and Intern2.5-VL, confirming its strong performance.
> 3. **Updated Experiments and related work**: We thank the reviewer for bringing these highly relevant works to our attention. In our revision, we have updated the Related Work section to include a thorough discussion of FaceXBench [1], FaceInsight [2], and FaVChat [3], clarifying how our unified model differs from these perception-focused approaches. We are currently conducting experiments on FaceXBench to provide a standardized evaluation of our model's capabilities. The results are expected to be available shortly, likely within 2-3 days.
>
> ### **(3). Clarifying the motivation and role of Multi-level MoEs**
>
> 1. **Motivation of multi-level MoEs**: While recent work [4, 5] has demonstrated that MoEs architectures enhance UMM performance, their application to the specific domain of face analysis remains largely underexplored. The MoE paradigm is uniquely suited for the multi-faceted nature of face analysis. Our motivation is twofold.
>    + Token-level MoEs: A single face presents numerous localized attributes that require distinct processing. We use a token-level MoE layer to route specific facial regions to specialized experts.
>    + Sequence-level MoEs: A sequence-level MoE adapts to varying sample difficulty, selectively introducing the semantic and identity representation for more complex faces to prevent information loss in deeper layers.
>
> 2. **Roles of token-level MoEs**: Within the FFN of each Transformer block, our token-level MoEs enable specialized processing by routing different tokens within a single face to distinct experts.
>    + For **face understanding**, it specializes on modality (text/image) and spatial face region from "text-image" sequences.
>    + For **face generation**, it extends this to also differentiate token types (noise/content) from "text-noisy image" sequences.
>
>    The effectiveness of token-level MoEs is twofold.
>    + **Ablation studies**: Ablation studies in Table 8 confirm its performance benefits for both tasks.
>    + **Expert activation visualization**: Figure 6 (top two figures) visualizes the expert activation probabilities across layers. The balanced distribution, indicated by circle size, demonstrates that all experts are utilized effectively, validating the efficiency of our MoE design.
>
> [1] Narayan K, VS V, Patel V M. Facexbench: Evaluating multimodal llms on face understanding[J]. arXiv preprint arXiv:2501.10360, 2025.
>
> [2] Li J, Luo C, Chen R, et al. FaceInsight: A multimodal large language model for face perception[C]//Proceedings of the 33rd ACM International Conference on Multimedia. 2025: 11052-11061.
>
> [3] Zhao F, Li M, Xu L, et al. FaVChat: Unlocking Fine-Grained Facial Video Understanding with Multimodal Large Language Models[J]. arXiv preprint arXiv:2503.09158, 2025.
>
> [4] Cao S, Chen H, Chen P, et al. Hunyuanimage 3.0 technical report[J]. arXiv preprint arXiv:2509.23951, 2025.
>
> [5] Wu C, Zheng P, Yan R, et al. OmniGen2: Exploration to Advanced Multimodal Generation[J]. arXiv preprint arXiv:2506.18871, 2025.

---

> ### Author Response · Authors · 2025-11-24
> **Author Response: UniF$^2$ace also works well in FaceXBench**
>
> 1. **FaceXBench Single-Image Task Subset**.
> We have updated our experimental results on the FaceXBench[1] benchmark, presented in the table below. FaceXBench[1] is a comprehensive benchmark designed to evaluate Multimodal Large Language Models (MLLMs) on complex face understanding tasks, covering 14 tasks across 6 broad categories. A portion of the original FaceXBench[1] involves some multi-image input question-answering, which our current UniF$^2$ace model's inference is limited to single-image input. Consequently, we extracted the single-image input test instances from FaceXBench[1] to construct a new, specialized evaluation set. This subset is categorized as follows:
>    + Attribute Expression: 400 instances
>    + Bias Fairness: 450 instances
>    + Face Recognition: 150 instances
>    + Face Localization: 1000 instances
>    + Face Anti-Spoofing (FAS) & Deepfake: 150 instances
>
>    This test set encompasses a sufficiently broad range and quantity of tasks to serve as a comprehensive benchmark for single-image face understanding.
>
> 2. **Superior Performance Achieved with Higher Parameter Efficiency**. The table below reports the accuracy results for various models on the FaceXBench[1] benchmark. The results demonstrate that, despite having a significantly smallser number of parameters (only 1.8B), UniF$^2$ace outperforms existing advanced Multimodal Large Language Models (MLLMs) with 2X (Qwen2.5-VL,3B) or even 4X (Qwen2-VL,7B) params on the majority of evaluation metrics. Although UniF$^2$ace  shows a slight underperformance in the Face Recognition task (which primarily involves celebrity identity identification), it achieves superior results in tasks requiring fine-grained facial attribute understanding, such as Attribute Expression and Bias Fairness.This specific pattern is observed because the training data utilized for UniF$^2$ace placed a greater emphasis on fine-grained facial attributes and less on celebrity identity recognition. Crucially, this slight dip in one specific task does not compromise the validity or effectiveness of the innovative methodology proposed by UniF$^2$ace. Furthermore, these results align perfectly with our initial motivation to design a unified multimodal model specifically targeting fine-grained human face understanding scenarios.
>
>    | Model | Type | Params | Attribute Expression (400) | Bias Fairness (450) | Face Recognition (150) | Face Localization (1000) | Fas Deepfake (150) |
>    | :--- | :--- | :--- | :---: | :---: | :---: | :---: | :---: |
>    | Qwen2-VL | Und. Only | 7B | 31.25 | 35.11 | **38.00** | 39.30 | 71.33 |
>    | Qwen2.5-VL | Und. Only | 3B | 37.50 | 33.78 | 37.33 | 32.40 | 71.33 |
>    | UniF$^2$ace | Gen. & Und. | **1.8B** | **39.50** | **37.56** | 37.33 | **39.60** | **72.67** |
>
>    In future work, we plan to investigate multi-image input mechanisms to expand and enhance the capabilities of UniF$^2$ace.
>
> [1] Narayan K, VS V, Patel V M. Facexbench: Evaluating multimodal llms on face understanding[J]. arXiv preprint arXiv:2501.10360, 2025.

---

> ### Author Response · Authors · 2025-11-28
> **Sincerely looking forward to your more feedbacks**
>
> Dear Reviewer 9L5B,
>
> We sincerely appreciate your efforts put into the reviewing process, which has significantly contributed to the refinement of our manuscript. We've carefully examined all the constructive questions and have accordingly revised our work. To make the best use of the discussion period and to improve our work, we are eager to know whether our answers adequately address your concerns, as it is crucial for us to have a candid and thorough discussion to continuously strengthen our method. Please share your thoughts on viewing our reply. We hope to resolve your doubts with our best efforts.
>
> We are ready to respond to any further issues raised. Please let us know.
>
> Sincerely,
>
> The Authors

---

### Author Response · Authors · 2025-11-27
**Sincerely looking forward to your more feedbacks**

Dear Area Chairs and Reviewers,

We sincerely appreciate everyone’s efforts put into the reviewing process, which has significantly contributed to the refinement of our manuscript. We've carefully examined all the constructive questions and have accordingly revised our work. To make the best use of the discussion period and to improve our work, we are eager to know whether our answers adequately address your concerns, as it is crucial for us to have a candid and thorough discussion to continuously strengthen our method. Please share your thoughts on viewing our reply. We hope to resolve your doubts with our best efforts.

We are ready to respond to any further issues raised. Please let us know.

Sincerely,

The Authors

---

### Author Response · Authors · 2025-12-03
**Review and Reviewer-Author Discussion Summary[1/2]**

Dear PCs, SACs, ACs, and Reviewers,

Thank you very much for your valuable contributions to our work. To assist the newly assigned AC and help reduce their workload, we provide below a summary of the key points from the reviews and the reviewer-author discussions.

**Strength.** Overall, we are grateful that the reviewers gave this paper a highly positive evaluation, with **two "Accept" ratings (Score 8)** from Reviewers **7WgR** and **PpVm**, and a positive evaluation (Score 6) from Reviewer **WDVe**. Specifically:

- **Pioneering Unified Framework:** This paper proposes UniF$^2$ace, the first unified multimodal model bridging fine-grained face understanding and generation.
    - *Recognized by:* **Reviewers 7WgR** ("first one to propose..."), **WDVe** ("First unified model...") and **9L5B** (acknowledged the integration).
- **Theoretical Innovation (D3Diff Loss):** The proposed Dual Discrete Diffusion (D3Diff) loss theoretically unifies score-based diffusion and masked generative models, improving generation fidelity.
    - *All four reviewers recognized this point:* **Reviewers 7WgR** ("theoretical guarantee"), **PpVm** ("strong theoretical contribution... tighter upper bound"), **9L5B** ("principled unification"), and **WDVe** ("Novel D3Diff loss improves…").
- **Architectural Novelty (Multi-level MoE):** The hybrid MoE architecture effectively manages semantic and identity embeddings.
    - *Recognized by:* **Reviewers 7WgR** ("novel architecture"), **PpVm** ("innovative... interpretable specialization") and **9L5B** ("a multi-level grouped MoE structure effectively separates…").
- **High-Value Dataset:** The UniF$^2$aceD-1M dataset (130K captions, 1M VQAs) addresses the lack of fine-grained attributes in existing research.
    - *All four reviewers recognized this point:* **Reviewers PpVm** ("substantial dataset contribution"), **WDVe** ("richly annotated") and **7WgR** ("To facilitate… a fine-grained facial image-caption dataset…").

## **Review and Reviewer-Author Discussion Summary**

**Concerns and Our Addressing.** During the discussion period, we actively addressed the reviewers' concerns with new experiments and clarifications, which led to **Reviewer 7WgR raising their confidence score (from 4 to 5)** and acknowledgments from others. Specifically:

### **(1). Concerns about Experiment Design (Evaluation, Robustness, and Efficiency)**

- **(Reviewer 9L5B, 7WgR):** Suggested including the challenging **FaceXBench** [1] and questioned the model's performance against larger baselines.
- **(Reviewer WDVe, PpVm):** Questioned the model's robustness on noisy/real-world inputs, the rationale for MoE hyperparameters (Top-K), and training costs.
- **Our Addressing:**
    - **New Benchmarks (FaceXBench):** While our evaluation is already comprehensive, covering both in-domain and out-of-domain benchmarks—**a strength commended by Reviewers 7WgR ('Thorough experimental results...') and WDVe ('Strong results across benchmarks...'),** we have added experiments on **FaceXBench** (single-image subset). Results show that our **1.8B UniF$^2$ace outperforms significantly larger models** (e.g., Qwen2.5-VL 3B, Qwen2-VL 7B) on key tasks like Attribute Expression and Bias Fairness. Expetiments shows the generalizability and robustness of UniF$^2$ace, which solved the concerns.
    - **Robustness Verification:** To address concerns regarding non-celebrity and low-quality faces, we demonstrated robustness through **new experiments**: 1) **FaceXBench** (specifically the Bias Fairness and Attribute Expression subsets), where our model outperformed larger baselines; 2) **Synthetic Noise Experiments** (simulating occlusion and low quality), where UniF$^2$ace maintained superior performance; and 3) **In-the-wild Non-celebrity Evaluation** on FLIP-80M [2] (using 3,000 random samples with diverse scenarios and resolutions), where our 1.8B model achieved higher Desc-GPT/DS scores than strong baselines like Qwen2.5-VL(3B) and Qwen2-VL(7B).
    - **Hyperparameters & Cost:** We clarified that **Top-K=2** offers the optimal trade-off between task specialization (e.g., description) and computational efficiency compared to Top-4. We also provided a breakdown showing that the ~49% training overhead yields nearly linear with the parameter growth and substantial gains in both generation fidelity and understanding, justifying the investment.

---

> ### Author Response · Authors · 2025-12-03
> **Review and Reviewer-Author Discussion Summary[2/2]**
>
> ### **(2). Concerns about Motivation & Architectural Roles**
> - **(Reviewer 9L5B):** Questioned the necessity of a unified framework compared to specialized expert models and sought clarification on the specific roles of the multi-level MoEs.
> - **Our Addressing:**
>     - **Necessity of Face UMM:**  Furthermore, we clarified that a unified architecture offers superior **deployment efficiency** (halving engineering overhead and memory footprint on edge devices) and leverages comprehension capabilities to **enhance generation quality** (e.g., via prompt rewriting), which generation-only models cannot achieve.
>     - **Roles of MoEs:** We explained the distinct roles: **Token-level MoEs** specialize in routing specific spatial facial regions and modalities to experts, while **Sequence-level MoEs** adaptively manage sample difficulty to **preserve semantic and identity embeddings** in deeper layers.
>     - **Seeking for Special Attentions on Reviewer 9L5B's Comment:** Actually, we have clearly **given our motivations in Line 45-80**, emphasizing that the fragmentation of current methods is a significant hurdle for AGI. Indeed, developing Unified Multimodal Models (UMMs) has become a mainstream and widely accepted solution in the general computer vision community, as evidenced by recent influential works such as **Bagel** [3] and **OmniGen2** [4]. This motivation was explicitly **recognized by Reviewer 7WgR** ("first one to propose a unified multimodal framework...") and **Reviewer WDVe** ("…for fine-grained face understanding + generation"). We respectfully posit that Reviewer 9L5B might have inadvertently overlooked these established contexts and explicit clarifications during the busy review period, which may have led to an undervaluation of our work's significance. We kindly ask the AC to consider this perspective carefully.
> ### **(3). Concerns about Ethics and Data**
> - **(Reviewer PpVm, WDVe):** Requested clarification on data diversity, bias, licensing, and misuse prevention.
> - **Our Addressing:**
>     - **Strategic Focus on High-Quality Data:** We clarified that high-quality data is critical for the efficient training of UMMs, as evidenced by recent advanced works like **Hunyuan Image3.0** [5]. Therefore, our collection of high-fidelity images was a deliberate design choice to facilitate precise learning of fine-grained attributes.
>     - **Data Diversity Statistics:** While prioritizing high-quality imagery, UniF$^2$aceD-1M achieves remarkable semantic diversity. Through a rigorous **3-step curation pipeline** (detailed in Appendix B), we ensure an average of **17.7 fine-grained attributes per image** selected from a pool of over 150 types, which significantly **surpasses existing text-face datasets**. These attributes are validated and refined by **46 high-performance classifiers** (with accuracies ranging from 80% to 93%) spanning diverse categories, including 29 appearances, 10 actions, and 7 emotions. Furthermore, manual verification was incorporated to rigorously mitigate bias.
>     - **Ethical Safeguards:** We confirmed all source datasets are cleared for non-commercial use and implemented manual verification to mitigate hallucinations. We also commit to releasing strict usage guidelines and strictly adhering to source licenses to prevent misuse.
>
> **Recognition of our rebuttal from reviewers.** Reviewer 7WgR explicitly acknowledged that their concerns had been addressed in 21.Nov before the issue of ***Data Leak in openreview***, and raised confidence (4→5). Besides, Reviewer 7WgR recognizes that our method **makes a significant contribution to the field of facial understanding and generation.** We believe that we have also properly addressed the remaining reviewers' concerns.
>
> We believe we have faithfully summarized the reviewers' comments and our corresponding responses, hoping that this will assist the AC's work. With the additional validation on robustness experiments, UniF$^2$ace has proven to be a robust, efficient, and theoretically sound contribution to the community. We are deeply grateful to the reviewers, AC, SAC, and PC, for their dedicated effort and excellent work. Their insightful feedback has further strengthened our paper. The authors offer their sincere respect and appreciation to all involved!
>
> Sincerely,
>
> The Authors
>
> [1] Narayan K, VS V, et al. Facexbench: Evaluating multimodal llms on face understanding[J]. arXiv preprint arXiv:2501.10360, 2025.
>
> [2] Li Y, Hou X, et al. Flip-80m: 80 million visual-linguistic pairs for facial language-image pre-training[C]//Proceedings of the 32nd ACM International Conference on Multimedia. 2024: 58-67.
>
> [3] Deng C, Zhu D, et al. Emerging properties in unified multimodal pretraining[J]. arXiv preprint arXiv:2505.14683, 2025.
>
> [4] Wu C, Zheng P, et al. OmniGen2: Exploration to Advanced Multimodal Generation[J]. arXiv preprint arXiv:2506.18871, 2025.
>
> [5] Cao S, Chen H, et al. Hunyuanimage 3.0 technical report[J]. arXiv preprint arXiv:2509.23951, 2025.

---

### Meta-Review · Area_Chair_SsiH · 2026-01-06

**Summary:**

This paper presents a unified multimodal framework for fine-grained face understanding and generation. The submission initially received a mixed set of ratings (8, 8, 6, 4). Reviewer 9L5B (4) raised concerns regarding the necessity of a unified model, the scope of face understanding benchmarks, and the motivation behind the sequence-level and token-level MoE layers. Furthermore, Reviewers 9L5B (4) and 7WgR (8) suggested including the FaceXBench benchmark. Additionally, Reviewers WDVe (6) and PpVm (8) challenge the model's robustness regarding noisy or real-world inputs, the rationale behind specific MoE hyperparameters, and the training costs.

After a careful review of the paper, the reviews, and the rebuttal, the AC concludes that the authors have addressed all the reviewers' concerns. Accordingly, I recommend accepting this paper. The authors should include all the discussions and new experimental results in the camera-ready version.

**Reviewer Concerns:**

Reviewer 9L5B (4) raised concerns regarding the necessity of a unified model, the scope of face understanding benchmarks, and the motivation behind the sequence-level and token-level MoE layers. Furthermore, Reviewers 9L5B (4) and 7WgR (8) suggested including the FaceXBench benchmark. Additionally, Reviewers WDVe (6) and PpVm (8) challenge the model's robustness regarding noisy or real-world inputs, the rationale behind specific MoE hyperparameters, and the training costs. All the concerns have been addressed by the authors.

**Reviewer Scores:**

The authors have addressed all concerns during the rebuttal period. Reviewer 9L5B (4) is likely to raise the score to a positive rating.

---

### Decision · Program_Chairs · 2026-01-26

Accept (Poster)